# Context information supports serial dependence of multiple visual objects across memory episodes

Cora Fischer [1], Stefan Czoschke [1], Benjamin Peters [1,2], Benjamin Rahm[3], Jochen Kaiser [1] & Christoph Bledowski [1 ✉]

Serial dependence is thought to promote perceptual stability by compensating for small changes of an object's appearance across memory episodes. So far, it has been studied in situations that comprised only a single object. The question of how we selectively create temporal stability of several objects remains unsolved. In a memory task, objects can be differentiated by their to-be-memorized feature (content) as well as accompanying discriminative features (context). We test whether congruent context features, in addition to content similarity, support serial dependence. In four experiments, we observe a stronger serial dependence between objects that share the same context features across trials. Apparently, the binding of content and context features is not erased but rather carried over to the subsequent memory episode. As this reflects temporal dependencies in natural settings, our findings reveal a mechanism that integrates corresponding content and context features to support stable representations of individualized objects over time.

[1] Institute of Medical Psychology, Goethe-University, Heinrich-Hoffmann-Strasse 10, 60528 Frankfurt am Main, Germany. [2] Zuckerman Mind Brain Behavior Institute, Columbia University, 3227 Broadway, New York, NY 10027, USA. [3] Medical Psychology and Medical Sociology, Faculty of Medicine, Albert-Ludwigs-University Freiburg, Rheinstraße 12, 79104 Freiburg, Germany. ✉email: bledowski@em.uni-frankfurt.de

Visual cognition relies on the interplay between perception and memory. We create visual objects by integrating features that belong together[1] and maintain object representations in visual working memory (WM)[2], which we can access flexibly when objects are no longer visible. Objects can slightly change their appearance from moment to moment due to movement or changed lighting, without changing their identity. As these changes of the world around us are often foreseeable, current object representations can be based on preceding ones. Thus, the exploitation of short-term dependencies represents an important requisite of experienced environmental stability.

A series of recent studies examined in detail how an object representation encoded in the previous trial influences an object representation currently encoded into WM. In their seminal study, Fischer and Whitney[3] found that the reported orientation in the current trial was systematically attracted by the orientation remembered in the previous trial. This bias was enhanced by spatial and temporal item proximity as well as by attentional selection and was termed 'serial dependence'. It has since been observed for other modalities, including facial identity and expression[4,5], spatial positions[6,7], numerosity[8–10] or ensemble representations[11]. As serial dependence reduces differences in the appearance of similar consecutive objects, it has been interpreted as promoting perceptual stability and continuity of a visual object over time[3].

If this interpretation was correct, which factors should support serial dependence? Following a single object over time requires some means of matching serial occurrences to decide whether the current object still represents the same or a novel object. This likely relies on matching occurrences according to similarities of their most relevant feature. Most previous studies examined serial dependence in a situation that comprised only a single relevant object per trial. In the real world, however, we are commonly confronted with situations where several objects have to be perceived and maintained in WM. In situations with multiple objects consisting of overlapping features, focusing on a single feature might not be sufficient. Instead, establishing object continuity should rely on the complete machinery that codes object identity in time and space, including coding of episodic regularities (e.g., serial position) and spatial positions as well as perceptual features like color that support object discrimination.

A parallel coding of different object features concurs with the idea that objects are maintained in WM as integrated representations of their features[1,2]. Such multi-feature representations would facilitate associations between objects over time. Alternatively, multi-feature objects could be represented in WM by a simultaneous, but independent storage of individual features[12]. Reconciling both approaches, Brady, Konkle and Alvarez[13] proposed that information in WM is structured as a hierarchical feature bundle consisting of two levels. The top level of a bundle represents an integrated object, while the bottom level contains independently stored low-level features. In line with this, Oberauer and Lin[14] proposed that objects in WM consist of several features that are bound together. They distinguished between object 'content' denoting the feature that needs to be reported, and object 'context' representing the feature dimensions via which an object can be cued for report. Context feature can hence refer to the spatial or serial position, but also to other object features like color. Moreover, context features can differ with regard to their relevance for the ongoing task. Context features that serve as cues to identify the currently task-relevant object can be defined as 'task-relevant context features' and context features that differ between objects but are neither reported nor serve as a cue can be defined as 'task-irrelevant context features'.

The present study tests a key prediction of serial dependence's interpretation as a mechanism that supports object continuity over time by investigating the role of context features. We hypothesize that when multiple objects are encoded into WM, they will be related to each other across trials via corresponding context features. This would indicate that context features, in combination with content features, leave traces in memory that support serial dependence between objects. Furthermore, as task-relevant context should be attended more than task-irrelevant context, the former should affect serial dependence more strongly. Finally, serial dependence should be influenced by whether or not a previous object served as a target that was selected and reported[15]. As attentional selection and retrieval processes affect the representations themselves[16–18], they are likely to influence the strength of the traces objects leave behind in WM. Accordingly, targets should produce a stronger serial dependence.

Indeed, we find that reported WM representations are biased toward previous representations, specifically to those that were targets and had corresponding task-relevant context features. This finding supports the idea of serial dependence as a mechanism that accomplishes object continuity by selectively integrating corresponding multi-feature representations of individualized objects.

## Results

**Behavioral paradigm**. In four experiments, subjects memorized two visual objects per trial. After a short delay, one of them was cued for report (Fig. 1). While motion direction served as the to-be-memorized content feature, objects were further distinguished by several context features. These context features were task-relevant when they served as a cue to indicate the target object, otherwise they were task-irrelevant. We estimated serial dependence, i.e., a systematic bias of the report error toward objects of the previous trial, between the cued object and each object from the previous trial with regard to their context congruence. In Experiment 1, two objects were presented sequentially with color and serial position as task-relevant and -irrelevant context feature, respectively. In Experiment 2, we reversed the task relevance of these context features. In Experiments 3 and 4 we used color and spatial position as task-relevant context features, respectively, and objects were presented simultaneously. Across the four experiments we studied the impact of three different context features: color, serial position and spatial position. These context features could have different impacts on serial dependence as they differ qualitatively. First, whereas color and spatial position are inherent to the visual presentation of an object, serial position arises solely in the temporal context of a memory episode with sequentially presented objects. Second, in contrast to color, temporal and spatial information are considered as fundamental for the definition of an object's identity[19].

**Experiment 1**. In line with previous findings[3,6,15,20], we observed significant serial dependence across trials even though two items were encoded per trial and one was cued for report. Specifically, the reported motion direction in the current trial was systematically attracted toward items presented in the previous trial. This attraction followed a derivative-of-Gaussian (DoG)-shaped curve with an amplitude parameter of 1.59° (bootstrapped SD: 0.303°, lower 95% of permutations between −1.38° and 0.89°, $p < 0.001$, $d_{est} = 0.966$, $R^2 = 0.078$) and a $w$ parameter of 0.033, equaling a width of 34.38° (full width at half maximum, FWHM). We ran one-sided permutation tests ($n = 20$) to assess whether this attraction differed from zero and between conditions.

Importantly, congruence of context features influenced serial dependence of object content (Fig. 2). Serial dependence was modulated by the task-relevant context feature, i.e., color, as it

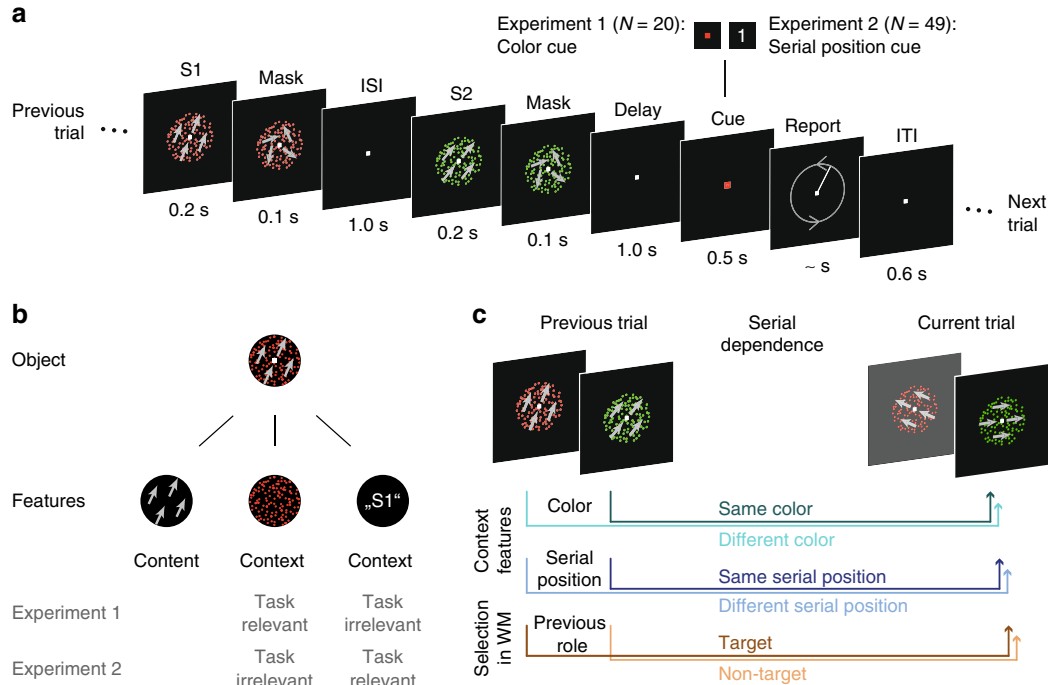

**Fig. 1 Experimental paradigm and object definition. a** In every trial, participants had to memorize motion directions (indicated here by gray arrows for illustration only) of two sequentially presented dot fields (S1 and S2) and report one of them after a short delay by adjusting the orientation of a line (possible adjustment directions indicated here by gray arrows for illustration only). The to-be-reported target item was cued via color (Experiment 1) or via serial position (Experiment 2). **b** Every object can be defined regarding its content feature, here motion direction of 25°, and regarding its context features, here color (red) and serial position (first in the sequence, S1). The context feature by which targets were cued was the task-relevant feature, whereas the other feature was task-irrelevant (e.g., color and serial position in Experiment 1, respectively). **c** We assessed the effects of three factors on the response error for a cued item (target) in the current trial: color (same or different), serial position (same or different) and role (target or non-target) of an item in the previous trial.

was stronger when the current item had the same color as a stimulus of the previous trial (amplitude = 2.32°, SD = 0.295°, lower 95% of permutations between −1.95° and 1.16°, $p < 0.001$, $d_{est} = 1.359$, $R^2 = 0.091$) as compared to when their colors differed (amplitude = 0.87°, SD = 0.646°, lower 95% of permutations between −1.30° and 0.84°, $p = 0.038$, $d_{est} = 0.252$, $R^2 = 0.012$) (amplitude difference = 1.46°, $p < 0.001$, $d_{est} = 0.856$,). Similarly, the observed serial dependence was more broadly tuned when the current item had the same color as a previous stimulus (FWHM = 37.45°) as compared to when they had different colors (FWHM = 28.59°) ($w$ difference: −0.010, equals FWHM difference: 8.86°, $p = 0.047$, $d_{est} = 0.727$). In contrast, task-irrelevant serial position modulated serial dependence only partially. The strength of serial dependence when the current stimulus was presented at the same serial position (amplitude = 1.56°, SD = 0.331°, lower 95% of permutations between −1.60° and 0.89°, $p < 0.001$, $d_{est} = 0.520$, $R^2 = 0.052$) was comparable to when they were presented at different serial positions (amplitude = 1.76°, SD = 0.303°, lower 95% of permutations between −1.43° and 0.94°, $p < 0.001$, $d_{est} = 1.178$, $R^2 = 0.046$) (amplitude difference = −0.20°, $p = .239$, $d_{est} = −0.328$). However, we observed a more broadly tuned serial dependence when current and previous stimuli were presented at the same serial position (FWHM = 39.45°) as compared to when their serial positions differed (FWHM = 28.03°) ($w$ difference: −0.012, equals FWHM difference: 11.42°, $p = 0.011$, $d_{est} = 0.266$).

Experiment 1 revealed a significant attractive bias only toward target items of the previous trial. We observed serial dependence on previous targets (amplitude = 2.99°, SD = 0.428°, lower 95% of permutations between −2.80° and 1.40°, $p < 0.001$, $d_{est} = 1.351$, $R^2 = 0.140$; FWHM = 38.30°), but not toward previous non-targets

(amplitude = 0.71°, SD = 0.699°, lower 95% of permutations between −1.23° and 0.72°, $p = 0.0519$, $d_{est} = 0.358$, $R^2 = 0.003$; FWHM = 11.33°). Therefore, the parameters of the curve-fitting analysis were not interpretable for the effect from non-targets. However, since the permutation test was nearly significant, we calculated the test between the two factors, which showed a stronger and more broadly tuned serial dependence from targets than from non-targets (amplitude difference = 2.28°, $p < 0.001$, $d_{est} = 1.042$; $w$ difference: −0.070, equals FWHM difference: 26.97°, $p < 0.001$, $d_{est} = 1.359$), but emphasize that this result should be interpreted cautiously.

In addition to the computed contrasts, we examined possible interactions between color, serial position and previous role, but found no significant results (details in Supplementary Fig. 1 and Supplementary Table 1). Furthermore, we computed a 4-way ANOVA with the factors color, previous item position, current item position and previous role (see Supplementary Fig. 2 and Supplementary Table 5).

**Experiment 2.** Congruence of color and serial position affected serial dependence differently in Experiment 1. This observation could not be unambiguously attributed to task relevance, because both features also differed in salience. While color is a salient feature inherent in the visual appearance of an item, serial position defines the temporal structure of a trial and is inherent in the encoding of the stimuli. To test whether the task relevance of a context feature determined the modulation by context congruence, we swapped the task relevance of both features in Experiment 2. We also eliminated any association between serial position and color by allowing two items within a trial to have the same color. Moreover, Experiment 1 only revealed a significant

Experiment 1: sequential presentation with color cue

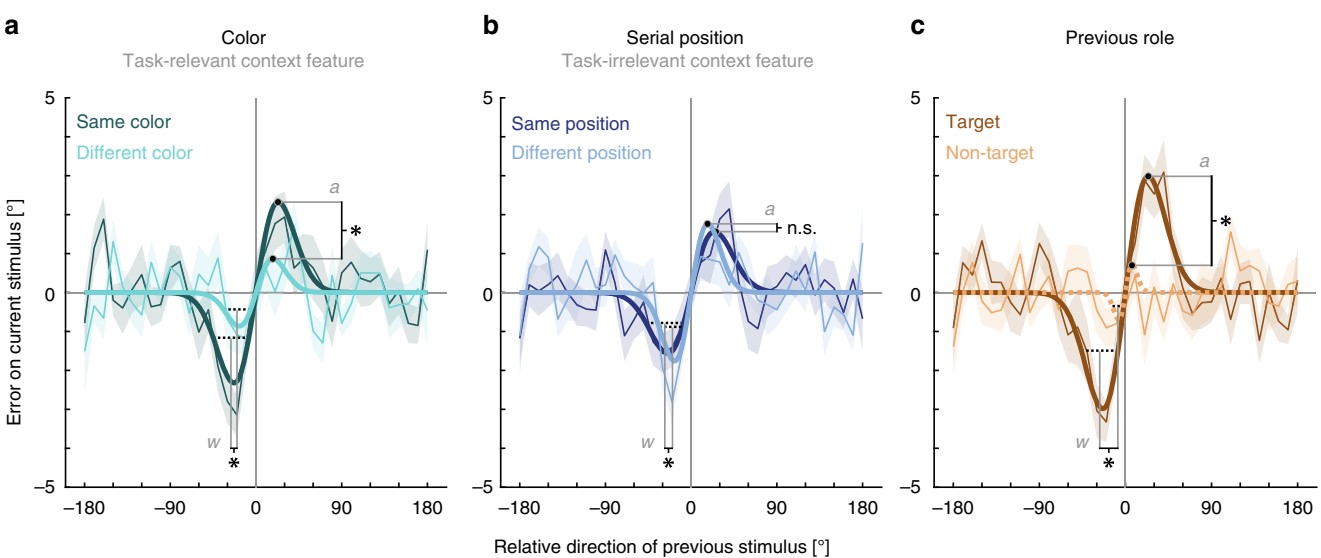

**Fig. 2 Results of Experiment 1.** The response errors (ordinate) are shown as a function of the motion direction difference (abscissa) between an item of the previous trial and the target of the current trial. Positive values on the abscissa indicate that the target direction was shifted counter-clockwise relative to an item of the previous trial. Positive values on the ordinate indicate that the response direction deviated clockwise from the true target direction. Serial dependence was revealed by the group averages of response errors (thin lines), with the corresponding shaded regions depicting the standard error of the group mean. A derivative of Gaussian (DoG, model fit shown as bold lines) was fitted to the response errors to estimate the systematic response bias. Solid lines indicate a significant bias, whereas dashed lines depict a non-significant bias Black filled circles and black dashed lines are showing the amplitudes and FWHMs of the DoG fits and are accompanied by a symbol indicating the result of the direct comparison of the parameter estimates (one-sided permutation tests, 20 participants). Asterisks indicate a $p$-value < 0.05 and the label 'n.s.' indicates a non-significant result with a $p$-value ≥ 0.05. **a** Both amplitude and width of serial dependence were greater between items with the same color than between items with different colors (amplitude difference: $p < 0.001$, width difference: $p = 0.047$). **b** The width of serial dependence was greater between items with the same serial position than between items with different serial positions (amplitude difference: $p = 0.239$, width difference: $p = 0.011$). **c** A significant serial dependence was observed from a target of the previous trial, but not from a previous non-target (amplitude difference: $p < 0.001$, width difference: $p < 0.001$). Source data are provided as a Source Data file.

attractive bias toward previous target items. To obtain more conclusive evidence about the potential serial dependence on non-targets, we substantially increased the number of participants.

We again observed serial dependence across trials when the target item was cued via serial position, with an amplitude of 2.00° (SD = 0.200°, lower 95% of permutations between −0.93° and 0.57°, $p < 0.001$, $d_{est} = 1.123$, $R^2 = 0.118$) and a FWHM of 34.63°. We ran one-sided permutation tests ($n = 49$) to assess whether this attraction differed from zero and between conditions.

Crucially, Experiment 2 resolved two open questions. First, it showed that task relevance determines the effect of context congruence on serial dependence. We found that serial dependence was modulated by the task-relevant context feature, i.e., serial position (Fig. 3). The strength of serial dependence was enhanced when the current stimulus was presented at the same serial position as a previous one (amplitude = 2.33°, SD = 0.204°, lower 95% of permutations between −1.15° and 0.71°, $p < 0.001$, $d_{est} = 1.243$, $R^2 = 0.106$) as compared to when they were presented at different serial positions (amplitude = 1.82°, SD = 0.200°, lower 95% of permutations between −0.87° and 0.56°, $p < 0.001$, $d_{est} = 0.881$, $R^2 = 0.045$) (amplitude difference = 0.52°, $p = 0.009$, $d_{est} = 0.387$). Additionally, we observed a more broadly tuned serial dependence between items at the same (FWHM = 44.60°) compared with different serial positions (FWHM = 26.04°) ($w$ difference: −0.018, equals FWHM difference: 18.56°, $p < 0.001$, $d_{est} = 0.840$). In contrast, color, which was task-relevant in Experiment 1 but task-irrelevant in Experiment 2, did not modulate serial dependence. Serial dependence when the current item had the same color as a previous stimulus (amplitude = 2.06°, SD = 0.195°, lower 95%

of permutations between −1.19° and 0.65°, $p < 0.001$, $d_{est} = 0.674$, $R^2 = 0.076$) was comparable to when their colors differed (amplitude = 1.93°, SD = 0.275°, lower 95% of permutations between −1.33° and 0.63°, $p < 0.001$, $d_{est} = 0.812$, $R^2 = 0.057$) (amplitude difference = 0.13°, $p = 0.347$, $d_{est} = −0.065$). The same was true for tuning widths (same color: FWHM = 36.00°, different colors: FWHM = 33.27°, $w$ difference = −0.0003, equals FWHM difference: 2.73°, $p = 0.228$, $d_{est} < 0.001$).

Second, Experiment 2 resolved the open question whether non-targets from previous trials can produce serial dependence, too. By increasing the number of subjects we indeed observed an attractive bias toward previous non-targets, albeit with a notably smaller amplitude for previous non-targets than targets (previous targets: amplitude = 3.46°, SD = 0.240°, lower 95% of permutations between −1.59° and 0.92°, $p < 0.001$, $d_{est} = 1.654$, $R^2 = 0.176$; previous non-targets: amplitude = 0.60°, SD = 0.297°, lower 95% of permutations between −0.62° and 0.44°, $p = 0.009$, $d_{est} = 0.421$, $R^2 = 0.005$; amplitude difference between previous targets and non-targets: 2.86°, $p < 0.001$, $d_{est} = 1.192$). This was also true for tuning widths (previous targets: FWHM = 37.98; previous non-targets: FWHM = 22.28°; $w$ difference = −0.021, equals FWHM difference: 15.70°, $p < 0.001$, $d_{est} = 1.070$).

We again examined possible interactions, which yielded no significant results (see Supplementary Fig. 1 and Supplementary Table 2) and computed a 4-way ANOVA (see Supplementary Fig. 2 and Supplementary Table 6).

**Experiments 3 and 4.** To generalize the results of Experiments 1 and 2 to the frequent situation of simultaneous object presentation, we presented two colored dot fields simultaneously at

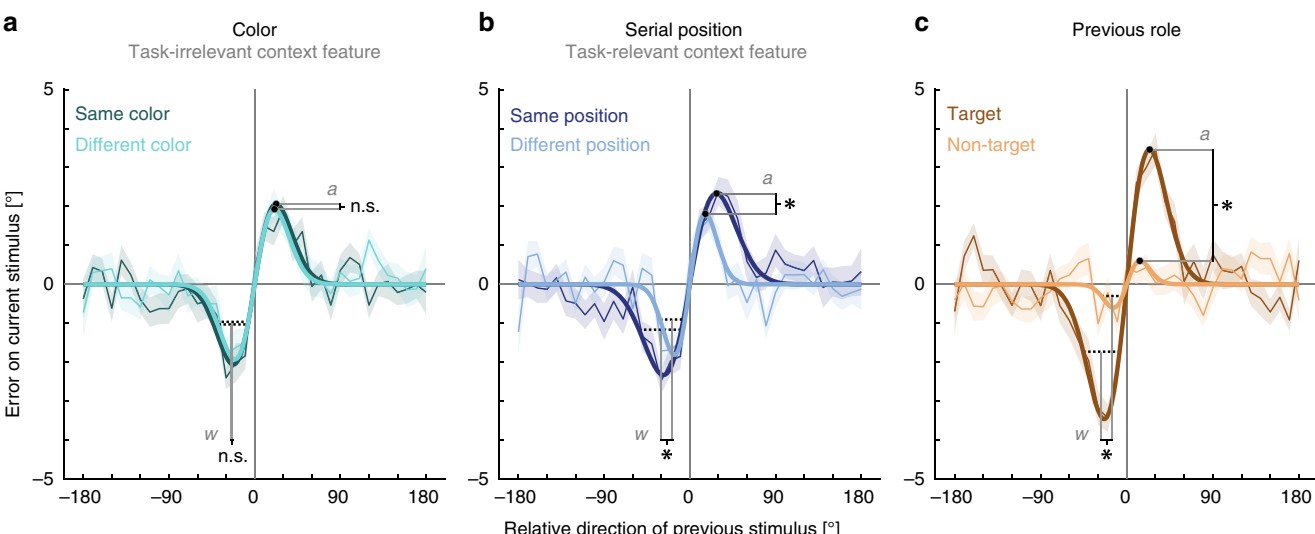

**Fig. 3 Results of Experiment 2.** Serial dependence (ordinate) is shown as a function of the motion direction difference (abscissa) between an item of the previous trial and the target of the current trial. For details see Fig. 2 and Methods. Serial dependence was revealed by the group averages of response errors (thin lines), with the corresponding shaded regions depicting the standard error of the group mean. A DoG model (shown as bold lines) was fitted to the response errors to estimate the systematic response bias. Solid lines indicate a significant bias, whereas dashed lines depict a non-significant bias. Black filled circles and black dashed lines are showing the amplitudes and FWHMs of the DoG fits and are accompanied by a symbol indicating the result of the direct comparison of the parameter estimates (one-sided permutation tests, 49 participants). Asterisks indicate a $p$-value < 0.05 and 'n.s.' indicates a non-significant result with a $p$-value ≥ 0.05. **a** Serial dependence did not differ significantly between items with the same color and different colors (amplitude difference: $p = 0.347$, width difference: $p = 0.228$). **b** Both amplitude and width of serial dependence were greater between items with the same serial position than between items with different serial positions (amplitude difference: $p = 0.009$, width difference: $p < 0.001$). **c** Both amplitude and width of serial dependence were greater for previous targets than non-targets (amplitude difference: $p < 0.001$, width difference: $p < 0.001$) Source data are provided as a Source Data file.

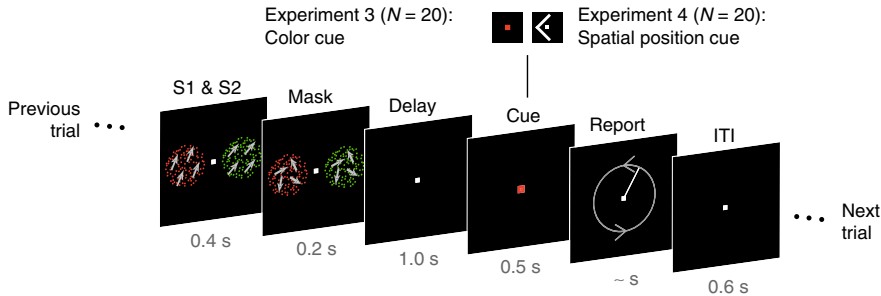

**Fig. 4 Experimental paradigm of Experiments 3 and 4.** In every trial, participants had to memorize motion directions (indicated here by gray arrows for illustration only) of two simultaneously presented dot fields (S1 and S2) and report one of them after a short delay by adjusting the orientation of a line (possible adjustment directions indicated here by gray arrows for illustration only). The to-be-reported target item was cued via color (Experiment 3) or via spatial position (Experiment 4).

different spatial positions (Fig. 4) in Experiments 3 and 4. In Experiment 3, spatial position was task-irrelevant and color was used as the cueing feature (analogously to Experiment 1), whereas in Experiment 4, spatial position was the task-relevant context feature.

Experiments 3 and 4 showed significant serial dependence across trials (Experiment 3: amplitude = 2.14°, SD = 0.383°, lower 95% of permutations between −1.98° and 1.18°, $p < 0.001$, $d_{est} = 0.551$, $R^2 = 0.042$ and FWHM = 40.33°; Experiment 4: amplitude = 1.51°, SD = 0.477°, lower 95% of permutations between −1.72° and 0.90°, $p < 0.001$, $d_{est} = 0.807$, $R^2 = 0.034$ and FWHM = 32.65°). We ran one-sided permutation tests ($n = 20$ in each experiment) to assess whether this attraction differed from zero and between conditions.

Importantly, Experiments 3 and 4 revealed that the results from Experiments 1 and 2 regarding the impact of the task-

relevant context features generalize to conditions with simultaneous presentation (Fig. 5). Specifically, in Experiment 3 we observed that task-relevant color affected serial dependence as shown by a significant attractive bias when the current stimulus had the same color as a previous one (amplitude = 2.13°, SD = 0.492°, lower 95% of permutations between −2.31° and 1.37°, $p < 0.001$, $d_{est} = 0.585$, $R^2 = 0.052$; FWHM = 45.66°) but not between stimuli with different colors (amplitude = −0.62°, SD = 0.991°, lower 95% of permutations between −1.77° and 1.08°, $p = 0.224$, $d_{est} = −0.806$, $R^2 = 0.011$; FWHM = 113.32°). The direct statistical comparison, which should be treated with caution, indicated a significant amplitude difference of 2.74° ($p = 0.005$, $d_{est} = 1.112$), but no significant difference between tuning widths ($w$ difference: 0.015, equals FWHM difference: −67.65°, $p = 0.153$, $d_{est} = 0.604$). Similarly, Experiment 4 also showed that task-relevant spatial position modulated serial dependence. We

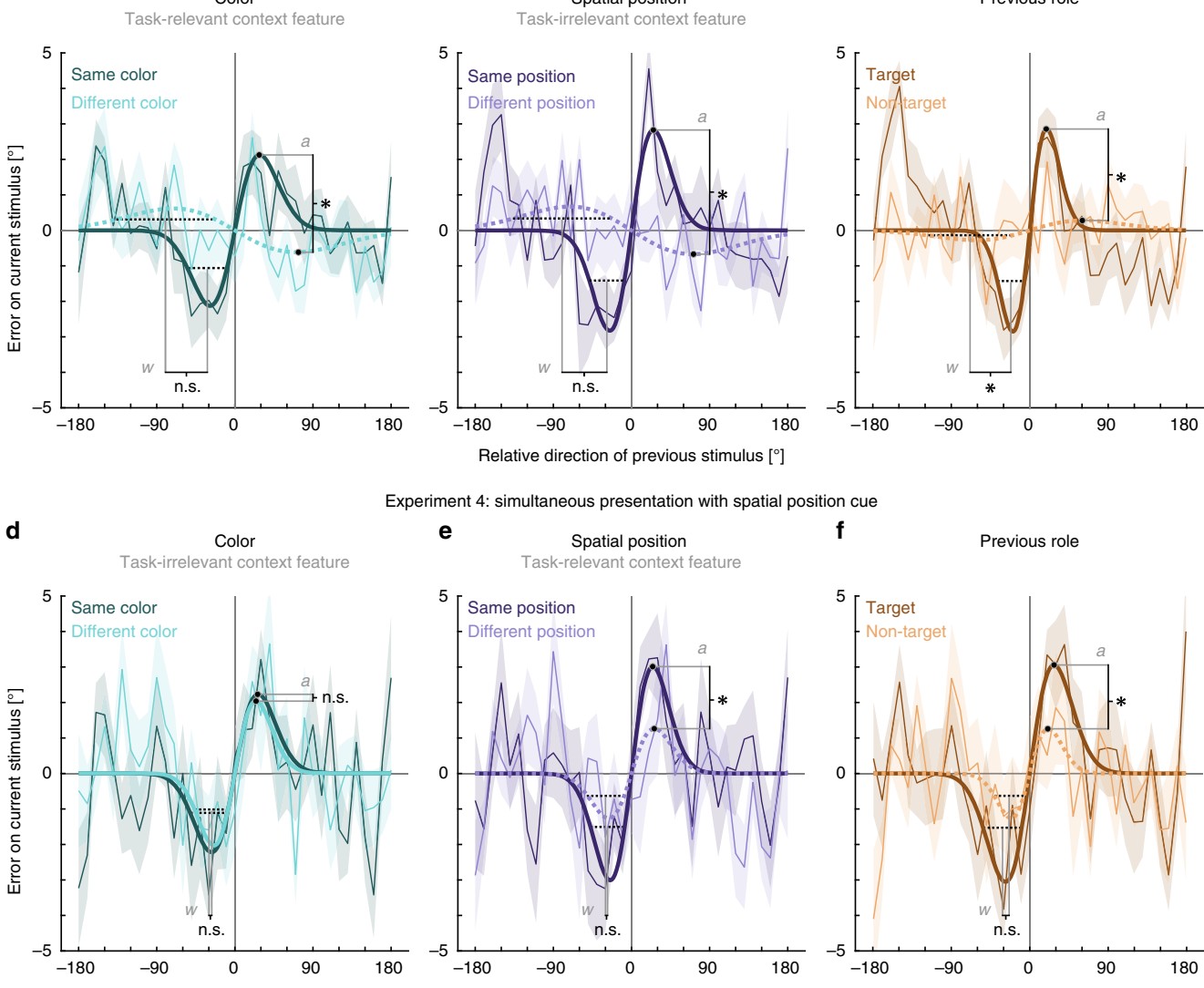

**Fig. 5 Results of Experiments 3 and 4.** Serial dependence (ordinate) is shown as a function of the motion direction difference (abscissa) between an item of the previous trial and the target of the current trial (Experiment 3: panels **a–c**, Experiment 4: panels **d–f**). For details, see Fig. 2 and Methods. Serial dependence was revealed by the group averages of response errors (thin lines), with the corresponding shaded regions depicting the standard error of the group mean. A DoG model (shown as bold lines) was fitted to the response errors to estimate the systematic response bias. Solid lines indicate a significant bias, whereas dashed lines depict a non-significant bias. Black filled circles and black dashed lines show the amplitudes and FWHMs of the DoG fits and are accompanied by a symbol indicating the result of the direct comparison of the parameter estimates (one-sided permutation tests, 20 participants per Experiment). Asterisks indicate a $p$-value < 0.05 and 'n.s.' indicates a non-significant result with a $p$-value $\geq$ 0.05. **a** Serial dependence was observed between items with the same color, but not between items with different colors (amplitude difference: $p = 0.005$, width difference: $p = 0.153$). **b** Serial dependence was observed between items with the same spatial position, but not between items with different spatial positions (amplitude difference: $p < 0.001$, width difference: $p = 0.154$). **c** Serial dependence was observed from a target of the previous trial, but not from a previous non-target (amplitude difference: $p = 0.003$, width difference: $p = 0.012$). **d** Serial dependence was observed both for items with the same color and different colors, with no difference between both conditions (amplitude difference: $p = 0.434$, width difference: $p = 0.284$). **e** Serial dependence was observed between items with the same spatial position, but not between items with different spatial positions. (amplitude difference: $p = 0.005$, width difference: $p = 0.379$). **f** Serial dependence was observed from a target of the previous trial, but not from a previous non-target (amplitude difference: $p = 0.010$, width difference: $p = 0.148$). Source data are provided as a Source Data file.

observed an attractive bias when the current stimulus was presented at the same spatial position as a previous one (amplitude = 3.01°, SD = 0.409°, lower 95% of permutations between −2.75° and 1.58°, $p < 0.001$, $d_{est} = 0.879$, $R^2 = 0.040$; FWHM = 39.41°) but not between stimuli with different spatial positions (amplitude = 1.26°, SD = 0.891°, lower 95% of permutations between −2.68° and 1.63°, $p = 0.137$, $d_{est} = 0.358$,

$R^2 = 0.010$; FWHM = 42.22°). The contrast, which again should be treated with caution, indicated a stronger serial dependence between items with the same spatial position as compared to items with different spatial positions (amplitude difference = 1.75°, $p = 0.005$, $d_{est} = 0.151$), but no significant difference of tuning widths ($w$ difference: 0.002, equals FWHM difference: −2.81°, $p = 0.379$, $d_{est} = 0.806$).

Strikingly, Experiment 3 also revealed that spatial position affected serial dependence even when it was task-irrelevant. This effect was comparable to Experiment 4, where spatial position was task-relevant. We observed a significant serial dependence between items with the same spatial position (amplitude = 2.83°, SD = 0.561°, lower 95% of permutations between −2.73° and 1.48°, $p = 0.001$, $d_{est} = 0.528$, $R^2 = 0.069$; FWHM = 40.42°) but not between stimuli with different spatial positions (amplitude = −0.67°, SD = 0.506°, lower 95% of permutations between −2.11° and 1.26°, $p = 0.228$, $d_{est} = −0.147$, $R^2 = 0.014$; FWHM = 113.32°). The direct statistical comparison suggested a significant difference between the amplitudes of the bias (amplitude difference = 3.50°, $p < 0.001$, $d_{est} = 0.670$), but no difference between the tuning widths ($w$ difference: 0.018, equals FWHM difference: −72.90°, $p = 0.154$, $d_{est} = 1.146$). In contrast, color as the task-irrelevant context feature in Experiment 4 did not influence serial dependence. This is in line with the results from Experiment 2, where color was task-irrelevant as well. Serial dependence when the current item had the same color as a previous stimulus (amplitude = 2.23°, SD = 0.539°, lower 95% of permutations between −2.90° and 1.64°, $p = 0.004$, $d_{est} = 1.369$, $R^2 = 0.029$) was comparable to when they had different colors (amplitude = 2.04°, SD = 0.671°, lower 95% of permutations between −2.32° and 1.48°, $p = 0.007$, $d_{est} = 0.494$, $R^2 = 0.021$) (amplitude difference = 0.18°, $p = 0.434$, $d_{est} = 0.528$). The same was true for tuning widths (same color: FWHM = 42.88°, different colors: FWHM = 38.78°, $w$ difference = −0.003, equals FWHM difference: 4.10°, $p = 0.284$, $d_{est} = 0.374$).

Together, spatial position affected serial dependence regardless of its task relevance, whereas color only had an impact when it was task-relevant.

The results of Experiments 3 and 4 also corresponded well to Experiments 1 and 2 regarding the effect of previous role. Experiment 3 showed a significant serial dependence from previous targets (amplitude = 2.86°, SD = 0.749°, lower 95% of permutations between −2.55° and 1.47°, $p < 0.001$, $d_{est} = 0.912$, $R^2 = 0.058$; FWHM = 31.17°), but not from previous non-targets (amplitude = 0.28°, SD = 0.792°, lower 95% of permutations between −1.54° and 1.16°, $p = 0.391$, $d_{est} = 0.112$, $R^2 = 0.002$; FWHM = 97.99°). The direct statistical comparison suggested a significant amplitude difference of 2.58° ($p = 0.003$, $d_{est} = 0.703$), and a significant difference regarding the tuning width ($w$ difference: 0.025, equals FWHM difference: −66.82°, $p = 0.012$, $d_{est} = 0.502$). We observed a similar pattern of results in Experiment 4: significant serial dependence from previous targets (amplitude = 3.06°, SD = 0.569°, lower 95% of permutations between −2.59° and 1.54°, $p < 0.001$, $d_{est} = 0.905$, $R^2 = 0.052$; FWHM = 44.379°), but not from previous non-targets (amplitude = 1.26°, SD = 0.902°, lower 95% of permutations between −2.51° and 1.54°, $p = 0.096$, $d_{est} = 0.188$, $R^2 = 0.008$; FWHM = 34.19°). The direct statistical comparison between targets and non-targets suggested a significant amplitude difference of 1.79° ($p = 0.010$, $d_{est} = 0.648$), but no significant difference in the tuning width ($w$ difference: −0.008, equals FWHM difference: 10.60°, $p = 0.148$, $d_{est} = 0.806$).

We again examined possible interactions, which yielded no significant results (see Supplementary Fig. 1 and Supplementary Tables 3 and 4).

## Discussion

In a series of four experiments, we tested the hypothesis that the correspondence of context features (color, serial, or spatial position) across memory episodes facilitates serial dependence in situations where several objects are processed. We observed a stronger serial dependence between items with the same task-relevant context features. Moreover, spatial position and, to some extent, serial position facilitated serial dependence even when they were task-irrelevant. Furthermore, the attractive bias was stronger toward the previous target item. Together, this supports our view that correspondence between content-context feature bundles supports the stability of multiple individualized objects over time.

In most previous studies on serial dependence, only one item per trial was encoded into WM. We observed serial dependence also when two items were encoded into WM, replicating recent results from our laboratory[15]. Importantly, we also replicated the finding that serial dependence is more pronounced for retro-cued target items of the previous trial. Retro-cueing implies that targets were "internally" selected within WM. We previously observed a similarly strong serial dependence on items that are retro-cued but not reported (omission of the response phase in a subset of trials) as on reported targets[15] (see also refs. [3,7]). This makes it unlikely that the report itself is responsible for the enhanced serial dependence on targets. Therefore, we assume that attentional selection of an object in WM causes the modulation. Also, previous studies found enhanced serial dependence for pre-cued targets, i.e., as a result of "external" selection during perception[3]. Therefore, both external and internal attention promote serial dependence. Interestingly, the present Experiment 2 with a larger number of subjects revealed serial dependence also for non-targets, albeit much weaker compared to targets. As both targets and non-targets from the previous trial were stored in WM but were irrelevant for the current trial, this result indicates that attentional prioritization in the previous trial strongly modulates serial dependence, without being a necessary precondition for its occurrence.

The key property of serial dependence is its selective operation between objects with similar contents, which has been observed for a range of different content features (orientation[3], faces[4], spatial position[6], numerosity[8−10], ensemble representations[11] or motion direction[15]). Based on this property, Fischer and Whitney[3] suggested that serial dependence reduces small differences between consecutive content features to support the impression of a coherent environment. The present study demonstrated that context features also leave traces in WM and help to relate corresponding objects across memory episodes. Specifically, we observed a stronger serial dependence between motion directions with the same color (Experiments 1 and 3), serial position (Experiment 2) or spatial position (Experiments 3 and 4).

The observed modulation of serial dependence by both content similarity and context congruence suggests that WM represents single features of an object as bound together to some degree. This is in line with frameworks assuming a WM organization with integrated multi-feature objects[2,13,14]. The definition of a WM item as a combination of content and context features corresponds closely to the concept of object files[1,19]. An object file contains different object features that form a temporary representation, enabling us to track an object over time. Object identity should remain stable and immune against small changes in object appearance, caused, e.g., by movement or changing lighting conditions. Fischer and Whitney[3] proposed serial dependence as a mechanism underlying object continuity. However, across memory episodes with multiple relevant objects, serial dependence would serve temporal integration only if it operated in an object-file fashion by relating corresponding bundles of content and context features. Our observation that context congruence of objects promotes serial dependence supports the interpretation that serial dependence is a mechanism suited for temporal integration of object representations across time.

Previous studies on serial dependence with one stimulus per trial have yielded ambiguous results with regard to context

features. A recent study[5] observed stronger serial dependence of emotional expressions between faces of the same gender than different genders, but no modulation by ethnicity. Apparently, with only one item per trial, the influence of context features is limited and only "strong" context features such as gender affect serial dependence. In contrast, the present design requiring the selective report of one out of two items per trial should have increased the binding between content and context features, thus enhancing the influence of context on serial dependence.

While task-relevant context features enhanced serial dependence, the effect of task-irrelevant context features was less consistent. Task-irrelevant spatial position clearly affected serial dependence, as there was only a significant bias between items with the same spatial position. Task-irrelevant serial position changed the tuning width of serial dependence, but not the amplitude (for a test of the interdependence of amplitude and width estimation see Supplementary Fig. 3 and Supplementary Note 1). In contrast, there was no effect for task-irrelevant color. This suggests that in contrast to color, spatial and serial position are more automatically integrated into an object representation, which reflects the importance of spatiotemporal information for object definitions[1]. Specifically, the serial position of an item defines the temporal structure of a trial, which might be particularly crucial in the case of sequentially presented items at the same spatial position[21]. The observed influence of spatial position is in line with previous studies demonstrating a spatial tuning of serial dependence[3,22–24].

Furthermore, we observed that task relevance modulated the amplitude of serial dependence differently. Serial dependence was suppressed between differently colored items when color was task-relevant, but not when it was task-irrelevant. In contrast, serial dependence between items of the same color was comparable regardless of whether color was task-relevant or task-irrelevant. Task relevance of serial position enhanced the amplitude of serial dependence between items with the same serial position compared to when this feature was task-irrelevant, whereas the bias for items with different serial positions was similar regardless of its task relevance. Task relevance thus enhanced the amplitude of serial dependence between items with the same serial position whereas it suppressed the amplitude between items of different colors. Finally, spatial position modulated serial dependence regardless of its task relevance, as in both Experiments 3 and 4 we found a significant bias only between items with the same spatial position. This further strengthens the interpretation that color, serial position and spatial position might play different roles for the definition of objects.

A large body of research has shown that previously memorized items can impact current reaction times and accuracies, a phenomenon known as proactive interference[25]. It has been observed for both verbal stimuli[26,27] and visual features like colors or shapes[28]. Proactive interference and serial dependence both describe effects of previous WM episodes on current ones. This led to the assumption that both could arise from the same mechanism and result from the same traces of once maintained items in WM[29]. Kiyonaga et al.[29] pointed out that proactive interference and serial dependence represent similar serial effects but are interpreted in opposing ways: proactive interference as a malfunction that has to be overcome and serial dependence as a beneficial temporal smoothing mechanism. Our results demonstrate that corresponding context features support serial dependence. Similarly, proactive interference is facilitated for items with the same spatial position[28] or serial position in serial recall tasks[30], which supports the assumption of a shared underlying mechanism. Important differences between both phenomena challenge this hypothesis. Proactive interference arises because an

item from a previous trial is mistakenly assigned to the current trial, whereas serial dependence describes an integration of a previous into current content information. Furthermore, proactive interference effects mostly stemmed from previously encoded but not tested items, whereas serial dependence most strongly arises from previous targets[3,15]. Taken together, proactive interference and serial dependence differ with regard to their conceptual explanation and whether they arise from previous targets or non-targets, but both describe an influence from past memory events on current ones, affected by context congruence. Therefore, more research is needed to clarify if those phenomena stem from the same underlying mechanism.

Until now, the processing stage at which serial dependence occurs has remained unclear. While some studies have suggested a perceptual stage[3,11,22,31,32], others have provided evidence for a memory- or decision-related process[6,20,33,34]. These seemingly contradictory results can be reconciled by assuming that serial dependence operates on multiple stages during object processing, as recently proposed[7]. Serial dependence might originate from the perception, memorization or decision on a previous object. Similarly, the current object processing might be affected at different stages. For example, serial dependence could originate from a decisional process and alter perceptual or memory representations. Our results indicate that when multiple objects were held in WM, serial dependence is driven by the memory representation of context features rather than their perceptual appearance, as the effect of context congruence was apparent despite strong visual interference between the presentation of current and previous objects. However, our study does not clarify at which stage context features affect the processing of a current object. Future studies should elucidate the mechanisms at different processing levels that lead to serial dependence and how they potentially interact.

Our study showed a bias of WM representations toward previous representations, specifically to those with corresponding task-relevant context features and previous targets. This provides an insight into object processing. Apparently, the binding of content and context features is not erased after a memory episode, but carried over into subsequent episodes. Our results point toward serial dependence as a mechanism that selectively integrates corresponding multi-feature object representations over time and therefore supports the temporal stability of individual objects.

## Methods

**Experiment 1: Subjects.** Twenty-two subjects who were recruited from the Goethe-University Frankfurt and the Fresenius University of Applied Sciences Frankfurt participated in Experiment 1. All subjects reported normal or corrected-to-normal vision. Two subjects aborted the experiment after the practice trials due to difficulties to perform the task. This resulted in a sample of 20 subjects (10 male), aged between 18 and 32 years (mean: 23.3 years). All subjects gave informed consent and received either a remuneration of € 10/h or course credit. The study was approved by the Ethics Committee of the Medical Faculty of the Goethe-University Frankfurt am Main and therefore complied with their ethical regulations.

**Experiment 1: Stimuli.** Random dot patterns (RDP) were presented centrally on the screen and consisted of 200 dots colored in red (RGB: [255, 0, 0]) or green (RGB: [0, 0, 255]) on a black (RGB: [0, 0, 0]) background. The dots were presented within an invisible circular aperture which had a radius of 7.5° of visual angle. The dots had a diameter of 0.15° of visual angle and were placed randomly within the circular aperture of the RDP at stimulus onset. The dots moved with a velocity of 3.5°/s and fully coherent in a direction randomly drawn from a pool of directions between 5° and 355° spaced 10° from one another, therefore avoiding cardinal directions. Dots reaching the edge of the aperture were repositioned randomly on the edge of the opposing side of the aperture, so that dot density was kept constant throughout the presentation. Throughout the whole experiment a white fixation square with a diagonal of 0.15° of visual angle was presented centrally on the screen, except for the cue presentation, when the fixation square changed its color to red or green to cue which item should be reported. The item was reported by

adjusting a randomly oriented line to match the reported direction. The response line was white, with a width of 0.6° and a length equaling the dot field radius. The starting point of the line was the fixation square and the end point could be altered so that the line could point in all possible directions.

**Experiment 1: Procedure**. Experiment 1 consisted of a delayed-estimation task, in which two sequentially presented motion directions had to be retained in memory, one of which after a short delay was cued for report (Fig. 1). Specifically, subjects saw two sequentially presented RDPs per trial (S1 and S2), each with a different motion direction and a different color, either red or green. Each trial began with the presentation of the first stimulus (S1) for 200 ms followed by a noise mask for 100 ms consisting of dots moving with 0% coherence (i.e., randomly) and of the same color as the preceding RDP. After a 1000 ms interval (ISI) the second stimulus (S2) and its noise mask were presented for 200 ms and 100 ms, respectively. Following a delay of 1000 ms, the fixation square changed its color to red or green for 500 ms, thereby cueing which motion direction had to be reported. Immediately after cue offset, a randomly oriented line was presented. Subjects had to report the motion direction by rotating the line via horizontal mouse movements. No time limit was given for the response and subjects were encouraged to work as precisely as possible. After adjusting the line direction, subjects had to confirm their response by pressing the left mouse button. If the entered direction differed more than 30° from the cued direction, another line pointing in the correct direction was presented for 500 ms as an error feedback (see ref. [35], for a similar procedure). At the end of each trial, a fixation screen of 600 ms was presented. Subjects were instructed to fixate the fixation square throughout the whole experiment.

In every trial, two different directions were presented, differing between 10° and 170°, equally spaced in steps of 10°, from one another. Over trials, the direction differences were balanced so that every possible direction difference occurred equally often. The order of the dot field colors was balanced across trials, so that both items were presented in both colors equally often, but in each trial the two stimuli had different colors from one another. In half of the trials, the red stimulus had to be reported, and in the other half the green one, which was balanced over encoding positions and direction difference combinations. The order of the trials was not balanced, therefore resulting in a different number of trials per subject and condition for the serial dependence analysis. Each item could be described with regard to its content (i.e., its motion direction), its context (i.e., its color and serial position) and its previous role (i.e., whether it was a target or a non-target) (see Fig. 1). Additionally, here the context feature color served as the task-relevant cueing feature, whereas the context feature serial position was task-irrelevant (see Experiment 2 below for the opposite assignment). The bias produced by an item from a previous trial on the report of the target item of the current trial could therefore be investigated with regard to three analysis factors with two levels each: item color (same or different color as the target of the next trial), serial position (same or different serial position as the target of the next trial) and previous role (target or non-target) (Fig. 1). Therefore, the impact of three factors on serial dependence was examined in a $2 \times 2 \times 2$ design. On average, 392.24 trials per factor level combination were analyzed in Experiment 1, with an average 10.6 trials per factor level combination and distance, and 393.46 and 10.6 trials in Experiment 2, respectively.

Every subject completed 1632 trials in two sessions on different days, lasting for ~2 h each (including instruction and practice trials). Each session was divided into eight blocks of 102 trials with self-paced breaks in between. After 45 minutes a general break was given, which was around the half of one session for most subjects. Up to three subjects completed the experiment in parallel in a dimly lit room, acoustically and visually shielded from one another. Subjects were seated at a viewing distance of ~50 cm from the display. MATLAB software with the Psychophysics Toolbox extensions[36,37] was used for stimulus generation and presentation. Three different LCD-monitors with a 4:3 display ratio and running with 60 Hz refresh rate were used.

**Experiment 1: Analysis**. Prior to the estimation of serial dependence, we excluded trials in which the response error is at least 3 SDs higher than the subject's mean response error, or in which the response time exceeded 20 s, indicating potential attentional lapses, accounting on average for 2.74% of trials in Experiment 1 and 2.47% in Experiment 2. We also excluded the first trial of each session as well as trials following a break, because in the analysis of serial dependence we were interested in the effect of the previous trial on the current one. We also demeaned the response errors by subtracting the overall mean response error of a participant from each individual response error to remove general individual response biases independent of serial dependence.

The evaluation of serial dependence was based on individual response errors, defined as the deviation between presented and entered direction. The errors were sorted regarding the difference between the target stimulus of the current trial and a stimulus from the previous trial as well as the relation of difference between the current item and the item of the previous trial (clockwise or counter-clockwise). For the target item of a current trial, the influences of two different items of the previous trial (S1 previous trial and S2 previous trial) were evaluated separately. The difference was computed by subtracting the direction of the current item from the direction of the item of the previous trial. Therefore, when the current item was

oriented more clockwise or more counter-clockwise, this resulted in a negatively or positively signed distance, respectively. A mean response error for a signed distance (distance × relation) deviating from 0 indicated a systematic response bias. When the sign of this systemic bias matched the sign of the distance between the directions, it indicated an attractive response bias. Conversely, an opposite sign of the systematic bias compared to the signed distance indicated a repulsive response bias.

Trials were then sorted according to their respective levels of the three analysis factors (see Procedure) before computing the mean response bias per signed distance for each factor level combination. Then we could examine the effects of the two context factors as well as task relevance on serial dependence (see Fig. 1). To assess the effect of those factors, we computed the mean response bias per factor level by averaging response biases of the appropriate conditions. For example, to examine the effect of color, we computed the mean of mean response biases of all conditions where items had the same color (regardless of serial position and task relevance) and different colors, which resulted in one mean response bias per level of the factor "color", signed distance and subject.

The individual mean response biases were used to evaluate the serial dependence per contrast level. We fitted the first derivative of a Gaussian curve (DoG; e.g., ref. [3]), a model which is usually used to describe serial dependence. The DoG, given by

$$y = xawce^{-(wx)^2}$$

was fitted to the pooled mean response biases of all subjects (similar to the procedure by ref. [20]) per factor level, i.e., one data point per subject and distance for the respective factor level. In the DoG, $x$ is the relative direction difference of two stimuli, $a$ is the amplitude of the curve peak, $w$ scales the curve width, and $c$ is the constant $\sqrt{2}/e^{-0.5}$. The $w$ parameter was constrained to a value range of .01 to .1. We optimized the log likelihoods of our curve fitting using Bayesian adaptive direct search (BADS; ref. [38]). BADS alternates between a series of fast, local Bayesian optimization steps and a systematic, slower exploration of a mesh grid. To estimate the variability of the parameters $a$ and $w$, we bootstrapped the DoG curve fit 1,000 times, sampling the data with replacement on each iteration, and computed the standard deviation of the resulting bootstrapped distributions of $a$ and $w$ (see ref. [3] for a similar procedure). For an easier interpretation of the $w$ parameter, we converted it into the full width at half maximum (FWHM).

**Experiment 1: Statistical analysis**. Permutation tests were used to assess effects on the group level. Specifically, to test whether there was a significant serial dependence, we randomly inverted the signs of each participant's mean response error. Subsequently, we fitted a new DoG model to the pooled group data and collected the resulting amplitude parameters $a$ in a permutation distribution. We repeated this permutation test 1000 times. As p-values we report the percentage of permutations that led to equal or higher values for $a$ than the one estimated for the empirical data. Based on previous findings[3,15,20], we expected a positive serial dependence and therefore the significance level was set to $\alpha = 0.05$ (one-sided permutation test). Significance of the model fit was assessed by a permutation test of the $a$ parameter only. The $w$ parameter was constrained to a range that excludes zero and zero would be the expected value of $w$ if the fitted data randomly fluctuated around zero without a systematic bias.

To test whether serial dependence differed significantly between factor levels, we applied the same procedure as above except that we randomly shuffled the labels of factor levels per participant. Thereby we generated a distribution of $a$ and $w$ differences, against which the $a$ and $w$ difference of the empirical data could be tested. Based on our hypotheses, we expected an enhancement of serial dependence, either in strength (i.e., higher amplitude) or width (i.e., broader tuning width) for same color or serial position in comparison to different color or serial position as well as for targets in comparison to non-targets. Therefore, the significance level was set to $\alpha = 0.05$ (one-sided permutation test). The p-value was given by the proportion of permuted differences whose values were equal or greater than the empirical difference.

To analyze possible interactions, we used a slightly different approach. As a non-significant bias would be difficult to interpret in an interaction, we followed the procedure suggested by[20]. Specifically, we collapsed the mean response error across clockwise and counter-clockwise motion direction differences between previous and current items per participant and then computed the mean biases for motion direction differences within a range of >0° and <60° (i.e., the bins from 10° −50°) separately for each participant and condition. To access the main effects and interactions of the factors color, serial position and previous role, we conducted a 3-way repeated-measures ANOVA at a significance level of $\alpha = 0.05$. In addition, to evaluate the evidence for the null hypothesis of an absence of interactions between the investigated factors, we computed Bayes factors (BF_incl) based on a Bayesian repeated-measures ANOVA (with BF_10). Bayes factors quantify the likelihood of a model with a single effect or interaction given the data relative to the likelihood of a model without a single effect or interaction. Therefore, a BF_incl > 1 indicates evidence for the data being more likely to occur under a model including those effects or interactions, rather than a model without them. In contrast, a BF_incl < 1 would support the H_0 of the data being more likely under a model without these effects or interactions. According to Kass and Raftery[39], BF_incl between 1/3 and 3 indicate inconclusive evidence. Therefore, we interpreted BF_incl > 3 as

supporting $H_1$ and $BF_{incl} < 1/3$ as supporting $H_0$. The Bayesian repeated-measures ANOVA was calculated with the JASP software (see Reporting Summary for details) using the default settings of the Bayes factor package[40]. Therefore, the Bayesian repeated-measures ANOVAs used a multivariate Cauchy prior with a fixed-effects scale factor of $r = 0.5$, and a random effects scale factor of $r = 1$ centered on zero.

**Experiment 1: Effect size calculation.** Our fitting procedure yielded group estimates for amplitude and width, whereas for the calculation of mean-based effect sizes, individual estimates are necessary. As there is no standard procedure for estimating the effect sizes for our analysis, we aimed at obtaining an approximation of the effect sizes that reflects the effects revealed by our permutation tests as good as possible. An estimation of the effect sizes analog to Cohen's d was calculated separately for the amplitude and width of the serial dependence, based on the obtained fittings for every condition. We used computations of the individual data informed by the group fitting to obtain individual estimates. Specifically, for amplitude, we calculated the individual mean response bias per factor level at the corresponding motion direction difference (current trial versus previous trial) that was closest to the obtained peak of the fitted curve. For example, if the fitting procedure yielded a $w$ parameter that indicated a curve peak at 22°, we obtained the individual response errors for 20° and −20°, inverted the response error for the negative distance and then averaged the two values. For the width of the curve, we calculated individual FWHM estimates. Therefore we first smoothed the individual mean response biases across all motion direction difference per condition using locally weighted non-parametric regression fitting (LOESS, implemented as the fLOESS MATLAB function, see Reporting Summary; analog to e.g., ref. [6]). We then calculated the FWHM of the smoothed function corresponding to its individual maximum peak between 0 and 60°. If the individual maximum was negative, FWHM was set to 0°. Based on those estimated individual amplitudes and FWHMs we calculated an estimated Cohen's d as effect size in the following way for the comparison against zero:

$$d_{est} = \frac{\mu_{amp}}{\sigma_{amp}}$$

where $\mu_{amp}$ is the mean of the individual amplitude estimates and $\sigma_{amp}$ their standard deviation. For the amplitude contrasts between factor levels, we calculated an estimated Cohen's d as

$$d_{est} = \frac{\mu_{amp_1} - \mu_{amp_2}}{\sigma_{amp_{pooled(1,2)}}}$$

with $\mu_{amp_{1/2}}$ as the mean amplitude for the first and second factor level, respectively and $\sigma_{amp_{pooled(1,2)}}$ the pooled standard deviation. To calculate the effect sizes for the width contrasts between factor levels, the same formula was used as for amplitude contrasts, but with $\mu_{FWHM_{1/2}}$ and $\sigma_{FWHM_{pooled(1,2)}}$, respectively.

**Experiment 1: Software.** All Analysis were performed with MATLAB 2018a and the following toolboxes/functions: Circular Statistics Toolbox[41], BADS[38], fLOESS and EzyFit (see Reporting Summary for details).

**Experiment 2: Subjects.** Fifty-one subjects who were recruited from the Goethe-University Frankfurt and the Fresenius University of Applied Sciences Frankfurt participated in Experiment 2, none of whom had participated in Experiment 1. All subjects reported normal or corrected-to-normal vision. Two subjects were excluded from the final analysis due to poor task performance (SD of report error >3 SDs of the sample mean). We thus included 49 subjects (19 male), aged between 18 and 33 years (mean: 23.8 years). All subjects gave informed consent and received either a remuneration of € 10/h or course credit. The study was approved by the Ethics Committee of the Medical Faculty of the Goethe-University Frankfurt am Main and therefore complied with their ethical regulations.

**Experiment 2: Procedure and stimuli.** The procedure and stimuli used equaled the ones described for Experiment 1. There were only two differences. First, cueing was now based on the encoding position of the stimuli instead of their color (Fig. 1). A number cue replaced the fixation square to indicate which one of the presented directions had to be reported. To ensure that stimulus color yielded no information about the encoding position of the stimulus, as a second change the stimuli presented in one trial could now have either the same or different colors. This resulted in four possible color combinations within a trial (red-red, red-green, green-green, green-red). Each of those color combinations occurred equally often. In half of the trials, the first presented stimulus had to be reported, and in the other half the second one, which was balanced over the different color and direction difference combinations. Three different monitors with a 4:3 display ratio were used, two running with 60 Hz and one with 75 Hz refresh rate, but with all stimulus parameters kept constant. Subjects were again seated at a viewing distance of ~50 cm.

**Experiment 2: Analysis.** All conducted analysis steps were the same as in Experiment 1.

**Experiment 3 and 4: Subjects.** Forty-one subjects were recruited from the Goethe-University Frankfurt, the Fresenius University of Applied Sciences Frankfurt, the University of Finance Wiesbaden and the Technical University Darmstadt, none of whom had participated in Experiment 1 or 2. All subjects reported normal or corrected-to-normal vision. Twenty-one subjects participated in Experiment 3. One subject was excluded from the final analysis due to poor task performance (SD of report error >3 SDs of the sample mean). We thus included 20 subjects (four male), aged between 19 and 30 years (mean: 22.9 years) in the final analysis of Experiment 3. Twenty subjects participated in Experiment 4 (seven male, aged between 19 and 29 years, mean: 23.15 years) and were included in the final analysis. All subjects gave informed consent and received either a remuneration of € 10/h or course credit. The study was approved by the Ethics Committee of the Medical Faculty of the Goethe-University Frankfurt am Main.

**Experiment 3 and 4: Stimuli.** RDPs were presented on the left and right side of the screen. The dots were presented within an invisible circular aperture which had a radius of 5° of visual angle, i.e., 2/3 of the radius used in Experiment 1 and 2. The centers of the RDPs were located 10° of visual angle to the left or right of the fixation square. In order to keep the direction information per stimulus comparable to Experiment 1 and 2, we reduced the number of dots per RDP to 67, which mirrored the reduction of the area of the RDPs from Experiment 1 and 2 to 3 and 4. All other stimulus parameters were the same as in Experiment 1 and 2.

**Experiment 3 and 4: Procedure.** Experiments 3 and 4 consisted of a delayed-estimation task, in which two simultaneously presented motion directions had to be retained in memory, one of which was cued for report after a short delay (Fig. 4). Specifically, subjects saw two simultaneously presented RDPs per trial (left and right), each with a different motion direction. In Experiment 3, cueing was based on the color of the stimuli. The fixation square changed its color to red or green to indicate which of the presented directions had to be reported as in Experiment 1. Therefore, the stimuli presented in one trial always had different colors, red and green. Whether the left or right stimulus was red (and the other one green) was balanced across trials. In half of the trials, the red stimulus had to be reported, and in the other half the green one, which was balanced over the different spatial positions and direction difference combinations. In Experiment 4, the target was cued via its spatial position by presenting a white arrow at the fixation square pointing to the left or the right of the screen. Similar to Experiment 2, the stimuli presented in one trial could have either the same or different colors to ensure that stimulus color provided no information about the spatial position of the stimulus. This resulted in four possible color combinations within a trial (red-red, red-green, green-green, green-red). Each of those color combinations occurred equally often. Each trial began with the simultaneous presentation of both stimuli (left and right) for 400 ms followed by a noise mask for 200 ms consisting of dots moving with 0% coherence (i.e., randomly) and of the same color as the preceding RDP. The stimulus durations were doubled in comparison to Experiments 1 and 2 to achieve similar encoding durations per stimulus with a simultaneous presentation. Following a delay of 1000 ms, the fixation square changed its color to red or green for 500 ms (Experiment 3) or a white arrow pointing to the left or right appeared at the fixation square for 500 ms (Experiment 4), thereby cueing which motion direction had to be reported. The rest of the procedure was the same as in Experiments 1 and 2.

**Experiment 3 and 4: Analysis.** All conducted analysis steps were the same as in Experiments 1 and 2.

**Statistics and reproducibility.** We conducted four separate experiments, three of which included partial replications of the previous ones.

**Reporting summary.** Further information on research design is available in the Nature Research Reporting Summary linked to this article.

## Data availability
Data are available via https://osf.io/azpwy. The source data underlying Figs. 2, 3 and 5, Supplementary Figs. 1–3 and Supplementary Tables 1–6 are provided as a Source Data File. A reporting summary for this Article is available as a Supplementary Information file.

## Code availability
Code is available via https://osf.io/azpwy.

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

## Acknowledgements
This study was supported by the German Academic Scholarship Foundation (PhD Scholarship awarded to C.F.). We thank Plamina Dimanova, Alina Rebitzky and Alicia Mohn for their help in data collection as well as Julia Krebs for help with piloting.

## Author contributions
C.F., S.C., B.P., B.R., J.K., and C.B. designed the experiments. C.F. conducted the experiments. C.F. analyzed the data. All authors contributed to the analysis approach and to data interpretation. C.F. and C.B. wrote the first version of the manuscript and all authors contributed in reviewing and editing.

## Competing interests
The authors declare no competing interests.
