## [Peer Review File · Nature Communications]

Reviewers' comments:

Reviewer #1 (Remarks to the Author):

Two clever experiments test whether context as well as target object content generates serial dependence in object representations. The experiment design is unique and novel, having pairs of stimuli in each trial, with contextual information (color or serial position) that is either task relevant or task irrelevant. The authors found that previous targets generate more serial dependence, and, interestingly, task relevant contextual information modulates the degree of serial dependence. The experiments are innovative and the writing is very clear, despite the fairly complex design. This is a very nice manuscript that will make an excellent addition to the literature. I only have minor comments, which do not dampen my enthusiasm at all.

1. There is a recent paper, relevant to the current study, by Liberman et al. (AP&P, 2018, "Serial dependence promotes the stability of perceived emotional expression depending on face similarity." It would be worth discussing this paper. It doesn't disagree or contradict the current findings, but it complements them because it was about a single object that had multiple dimensions.
2. Line 645 and Discussion section. "the present results support the ...memory-based or decisional..." The discussion section could be strengthened enormously by avoiding this simplistic statement here and instead having a deeper discussion about the likelihood that serial dependence is not one grand unified thing but actually occurs at many stages. Kiyonaga et al suggested this in 2017 and the present results are much more consistent with that review, and the idea of serial dependencies at multiple stages, rather than the oversimplified idea that serial dependence can "only" happen either in perception or in memory/decision. While lines 640-44 are ok, insofar as much of the previous literature was couched in the (oversimplified) black and white distinction, the present manuscript is the ideal time to start moving away from the strawman distinction and instead discuss the more nuanced idea that serial dependence can operate at multiple levels.
3. Related to number 2. The current results do not actually test or rule out the role of serial dependence in perceptual appearance. The previous (positive) demonstrations of serial dependence in appearance remain as existence proof (Fischer & Whitney, 2014 supplemental material; Cicchini et al., 2017; Fornaciai et al; Manassi et al). Those results are not overturned by the present results. I do agree with the authors that the present experiments "argue against an exclusively perceptual basis of serial dependence". However, I don't think most papers argue for that. For example, Manassi et al, 2018, PB&R (which should be cited) stated that:
"It is also possible that there is serial dependence at multiple levels of representation, including at basic perceptual levels (Cicchini et al., 2017; Fischer & Whitney, 2014), perceptual decisions (Fritsche et al., 2017), and also in memory representations (Papadimitriou et al., 2015; Zhang, Liberman, & Whitney, 2016)." (discussion section, Manassi et al., 2018, PB&R)
4. Line 75. Add references to the numerosity serial dependence literature here in the list (Cicchini, et al PNAS, 2014; Corbett et al., PLoS one, 2011; Fornaciai and Park, 2018, JoV).

Reviewer #2 (Remarks to the Author):

Dear All,

I have read with much attention the paper by Fischer et al as it is a submission for Nature Communication and, besides technical quality one has to foresee the possible impact of the paper itself.

In this respect let me briefly summarize the state of the art in the serial dependence literature which is the target of the current research. Serial dependence (attraction towards the previous stimulus) is a relatively new phenomenon and literature is growing fast and disorderly. At present it is not known if the effect of previous presentations affects what we see or what we respond. Neither we know if it is necessary to attend a stimulus in order to have an effect on the following trial. Also we don't know up to which degree we have to store some information in working memory in order to be biased on the subsequent trial. All of the aforementioned questions are still open as no conclusive answer has been reached however they have all been tackled to some extent by current literature. This creates a state of affairs in which, if one seeks to make a significant contribution, he/she has the duty to relate to previous findings and provide encompassing answers.

In this respect the current manuscript falls short. Obviously, the paper provides new results on serial effects. In particular the paper demonstrates that only the item out of two which was maintained in working memory will exert a conspicuous effect on the subsequent trial. Which I find quite intriguing. However the way the paper is drafted and the way research has been organized make the current manuscript far from exhaustive. Indeed the same group has demonstrated with a similar paradigm that if the subject is not requested to make a response on the previous trial there is still a conspicuous serial effect. What has a reader to conclude from both findings? One possibility is that previous stimuli leave active traces by default and it is only when resources are devoted to the other item (which is what happens to the non-targets) the visual trace is wiped out. Shouldn't the authors tackle this issue upfront? And in any event doesn't this observation somehow undermine the generality of the current findings?

At the same time it is worth of notice that "attractive" interferences in working memory have already been demonstrated by an enormous amount of literature. Sometimes, when there is a change of items in working memory the end result is interference (indeed the phenomenon is dubbed "proactive interference"). But in other paradigms, when two subsequent datasets contain more similar items intrusions of past stimuli does take place. Indeed in a recent TICS review Kyonaga et al have underlined the similarity between working memory effects and serial effects and suggested that they operate in a similar fashion. Again this pre-empties the interest on the current paper. Had the authors tackled the issue upfront, again, perhaps one would have an answer to a recent hypothesis which is receiving much attention. But the authors haven't.

The manuscript is also not meeting entirely its own expectations. The concept of content and context in

memory is presented in the introduction and even makes its way to the title. Indeed the paper promises to investigate whether context (i.e. the dimension which the experimenter will use to inform the subject upon which stimulus to judge) plays a role in serial effects. But the results are not that clear cut. In Experiment 1 where it the dimension indicating which stimulus to reproduce is color, and if the subsequent trial has the same color as the previous stimulus the effect is way larger than when the two stimuli have different colors. But in Experiment 2 the crucial dimension is “position in time” and then the extra serial effect when relating items in the same temporal position is only 20%. Does this support the authors claims? Why did they chose to study “position in time”? Why didn’t they worked along the “dot-size” dimension?

Another quite serious drawback of the manuscript is that several results have not been displayed clearly to the authors with graphs and are only described via the result of statistics. This makes it very difficult to gauge and absorb the effects at play. For instance the authors state that there are no interaction as for the amplitude of serial effects in Experiment 1 between the various dimensions. Does this mean that when one considers a previous stimulus which was a “non target” and had a “different color” one has a repulsive effect? Shouldn’t the readers have a direct representation of all such conditions? Thus I encourage (whichever the destiny of the current submission) to display all the sub-conditions perhaps in the form of bar plots, obviously separating amplitudes and FWHM.

A related point is that some stats (like those for the interaction) are entirely missing. Also it would help to incorporate Bayesian tests. This is particularly crucial as sometimes the authors suggest there is evidence in favour of the null hypothesis (like in the aforementioned interaction)

Finally the manuscript presents a rather narrow set of data and it is not trivial to generalize to other, similarly important, conditions. For instance why didn’t the authors test if this pattern of results would be identical if the two items were displayed in different positions? This variable is crucial in working memory and it is important to test if it plays the same role for serial dependence. Also why didn’t the authors split the data according to the sheer order of presentation in the previous trial? In other words, does the last stimulus of previous trial exert a stronger influence?

For all these major issues I think that the current manuscript is far from ready for a premier journal like Nature Communications. Somehow the findings are potentially interesting but there is a lot to do to become a reference for future scholars.

Having said this I would like also to mention two minor points.

The first is that the reader should be reassured that the estimates of FWHM are independent from the amplitude estimates. How robust is their fitting procedure? Isn’t it somehow a necessity that high amplitude effects can win over experimental noise and will be captured by a fitting function into their full extent whereas low amplitude effects will only be clear over a smaller range, then noise swamps the effect?

The second is the statement what the current findings speak against the perceptual interpretation of serial effects. Under some respect they do as they provide a case in which the mere presentation of a stimulus is not sufficient to yield an attractive effect. But at the same time this very paradigm (double presentation in the same position) is a bit peculiar and, as mentioned before, may not be that general. Even accepting that caching stimuli in working memory is crucial, still it is possible that the end effect is a warping of perception.

We thank the reviewers for their thorough and thoughtful reviews and their constructive criticism. We have carefully addressed each of their points during the revision of the manuscript. Importantly, in response to the remarks of reviewer #2, we conducted two additional experiments showing that our results generalized to the simultaneous presentation of visual objects that is often used in perception and working memory research. We think that the revision of our paper has substantially improved its quality and strengthened the empirical basis of its proposed wide scope.

Reviewers' comments:

Reviewer #1 (Remarks to the Author):

Two clever experiments test whether context as well as target object content generates serial dependence in object representations. The experiment design is unique and novel, having pairs of stimuli in each trial, with contextual information (color or serial position) that is either task relevant or task irrelevant. The authors found that previous targets generate more serial dependence, and, interestingly, task relevant contextual information modulates the degree of serial dependence. The experiments are innovative and the writing is very clear, despite the fairly complex design. This is a very nice manuscript that will make an excellent addition to the literature. I only have minor comments, which do not dampen my enthusiasm at all.

We are grateful to the reviewer for this encouraging response and her/his appreciation of our study.

1. There is a recent paper, relevant to the current study, by Liberman et al. (AP&P, 2018, "Serial dependence promotes the stability of perceived emotional expression depending on face similarity." It would be worth discussing this paper. It doesn't disagree or contradict the current findings, but it complements them because it was about a single object that had multiple dimensions.

We thank the reviewer for pointing us toward this work, which is indeed relevant for our study. Liberman et al. (2018) observed a modulation of serial dependence for emotional expression by face identity, which is in line with our results. We included the study in the discussion (line 863).

2. Line 645 and Discussion section. "the present results support the ...memory-based or decisional..." The discussion section could be strengthened enormously by avoiding this simplistic statement here and instead having a deeper discussion about the likelihood that serial dependence is not one grand unified thing but actually occurs at many stages. Kiyonaga et al suggested this in 2017 and the present results are much more consistent with that review, and the idea of serial dependencies at multiple stages, rather than the oversimplified idea that serial dependence can "only" happen either in perception or in memory/decision. While lines 640-44 are ok, insofar as much of the previous literature was couched in the (oversimplified) black and white distinction, the present manuscript is the ideal time to start moving away from the strawman distinction and instead discuss the more nuanced idea that serial dependence can operate at multiple levels.

We agree with the reviewer that the original discussion section was kept too simple and we now have substantially extended it. We especially concord with the viewpoint that serial dependence might arise from different processing levels of the previous object including perception, memorization or decision. We also note that the processing of the current object might be affected at different stages as well. While our results indicate that the attractive bias is based on memory representations of content-context bindings, they do not reveal at which stage of the processing of a current object the attraction takes place. We have substantially edited the related paragraph in the Discussion (starting from line 959).

3. Related to number 2. The current results do not actually test or rule out the role of serial dependence in perceptual appearance. The previous (positive) demonstrations of serial dependence in appearance remain as existence proof (Fischer & Whitney, 2014 supplemental material; Cicchini et al., 2017; Fornaciai et al; Manassi et al). Those results are not overturned by the present results. I do agree with the authors that the present experiments “argue against an exclusively perceptual basis of serial dependence”. However, I don’t think most papers argue for that. For example, Manassi et al, 2018, PB&R (which should be cited) stated that: “It is also possible that there is serial dependence at multiple levels of representation, including at basic perceptual levels (Cicchini et al., 2017; Fischer & Whitney, 2014), perceptual decisions (Fritsche et al., 2017), and also in memory representations (Papadimitriou et al., 2015; Zhang, Liberman, & Whitney, 2016).” (discussion section, Manassi et al., 2018, PB&R)

We agree with the reviewer that our study did not rule out that serial dependence may operate on the perceptual level. However, the present finding of a context modulation of serial dependence, cannot be explained by a purely perceptual account of the bias. But, as the reviewer pointed out, studies that found evidence for serial dependence on the perceptual level do not necessarily state that other processing levels are not involved. Furthermore, serial dependence could affect perceptual representations even if the effect originated from memory or decisional levels. We have changed the related paragraph in the Discussion (starting from line 959) accordingly and added the reference to Manassi et al. (2018).

4. Line 75. Add references to the numerosity serial dependence literature here in the list (Cicchini, et al PNAS, 2014; Corbett et al., PLoS one, 2011; Fornaciai and Park, 2018, JoV).

Thanks for pointing out this omission; we have added numerosity to the list of investigated modalities in the introduction (line 79) and the discussion (line 831).

Reviewer #2 (Remarks to the Author):

Dear All,

I have read with much attention the paper by Fischer et al as it is a submission for Nature Communication and, besides technical quality one has to foresee the possible impact of the paper itself.

1) In this respect let me briefly summarize the state of the art in the serial dependence literature which is the target of the current research. Serial dependence (attraction towards the previous stimulus) is a relatively new phenomenon and literature is growing fast and disorderly. At present it is not known if the effect of previous presentations affects what we see or what we respond. Neither we know if it is necessary to attend a stimulus in order to have an effect on the following trial.

Also we don't know up to which degree we have to store some information in working memory in order to be biased on the subsequent trial. All of the aforementioned questions are still open as no conclusive answer has been reached however they have all been tackled to some extent by current literature. This creates a state of affairs in which, if one seeks to make a significant contribution, he/she has the duty to relate to previous findings and provide encompassing answers.

We completely agree with the reviewer that the literature on serial dependence has grown rapidly, including an increasing number of clever and detailed studies aiming at specifying that phenomenon. Without a doubt, these have left open several relevant questions as listed by the reviewer. We feel that the present study addressed an equally important issue by assessing a fundamental assumption concerning the functional significance of serial dependence. According to the "continuity field" interpretation, serial dependence reflects a mechanism that promotes object stability over time by reducing small differences between objects. *If this commonly held interpretation were correct, which factors should support serial dependence and modulate the strength of the bias?* Following a single object over time requires some means of matching serial occurrences to decide whether the current object still represents the same or a novel object. This likely relies on matching occurrences according to their similarity with respect to their most relevant (target) feature. This has been the core of most previous research on serial dependence. However, in situations where multiple objects are present that may have overlapping features, focusing on a single feature might not be sufficient. We therefore expect that a mechanism promoting continuity and object stability relies on the complete machinery that codes object identity in time and space, including coding of episodic regularities (e.g. serial position) and spatial positions, and object features like its color that might appear irrelevant but do support object discrimination.

We thus feel that testing a key prediction of the assumption of serial dependence as a mechanism supporting object continuity over time by investigating the role of context effects makes a highly valuable contribution to the existing knowledge. We rewrote both the Introduction (starting from line 86) and Discussion (starting from line 787) to make the motivation and relevance of our study clearer.

Beyond this central question, we believe that our study also contributes to the open questions raised by the reviewer. Regarding the question at which level serial dependence operates, our results provide evidence that context features like serial position represent mnemonic rather than perceptual effects (see also our reply to comment 10). Regarding the role of attention, we showed that attentional selection of an object for report crucially enhances serial dependence,

but that it is not a necessary precondition (see also reply to comments 2 and 3). We have now included a new section in the Discussion, where we relate our findings more clearly to the question of the stage at which serial dependence occurs (starting from line 959). Also we discuss in greater detail how attention influences serial dependence (starting from line 808) and we have extended our discussion of the relationship between proactive interference and serial dependence (starting from line 913).

2) In this respect the current manuscript falls short. Obviously, the paper provides new results on serial effects. In particular the paper demonstrates that only the item out of two which was maintained in working memory will exert a conspicuous effect on the subsequent trial. Which I find quite intriguing.

We appreciate that the reviewer finds the difference between targets and non-targets interesting. However, we would like to emphasize that in our paradigm, both items in each trial had to be maintained in working memory. Only after a delay, one of them was retro-cued, hence became the target of a subsequent report (continuous recall), while the other item became task-irrelevant (non-target). Importantly, after report also the target became task-irrelevant. Thus, effects in serial dependence in our study always referred to the impact of items that were previously maintained in memory but were no longer task-relevant after the end of the trial (as in previous serial dependence studies). The crucial difference between the previous items was whether or not they served as targets. We found that in particular targets lead to clear and strong serial dependence, while the effect of non-targets was much smaller and only observable with sufficient statistical power (Experiment 2 with 49 subjects). In our previous study (Czoschke et al., 2019; Experiment 2), omitting the report phase in a subset of trials demonstrated that this effect did not depend on the report. This was in accordance with Experiment 2 in the original study by Fischer and Whitney (2014), who also found that serial dependence, was not a result of hysteresis in report. Moreover, Experiment 4 by Fischer and Whitney (2014) showed that only an item that was pre-cued influenced subsequent item processing. Taken together, these results suggest that in particular an object that was in the focus of attention at the end of the previous trial causes serial dependence. Apparently, this can be observed regardless of whether the item was selected among others during perception via a pre-cue as shown by previous research (Fischer & Whitney, 2014; Fritsche et al., 2017), or selected from memory via a retro-cue as in the present study.

3) However the way the paper is drafted and the way research has been organized make the current manuscript far from exhaustive. Indeed the same group has demonstrated with a similar paradigm that if the subject is not requested to make a response on the previous trial there is still a conspicuous serial effect. What has a reader to conclude from both findings? One possibility is that previous stimuli leave active traces by default and it is only when resources are devoted to the other item (which is what happens to the non-targets) the visual trace is wiped out. Shouldn't the authors tackle this issue upfront? And in any event doesn't this observation somehow undermine the generality of the current findings?

We thank the reviewer for pointing out that we should have related our current findings more explicitly to our previous ones. As described in our reply to the previous comment, we think that our previous findings in combination with our current set of results strengthens the interpretation that attentional selection amplifies serial dependence. We agree with the

reviewer that previous stimuli leave active traces by default and that those traces are especially impactful when the stimulus was previously internally selected. As mentioned before, we also observed a small serial dependence effect from the previous non-target in Experiment 2, which indicates that its trace was not completely wiped out. We regret not having made this relation clearer in the original submission. We have changed the Discussion accordingly (starting from line 808).

Regarding the point of whether the attentional modulation undermines the generality of our findings, we would like to point toward our findings showing that the influence of context features on serial dependence did not differ significantly between targets and non-targets. Specifically, across four experiments, we consistently observed main effects of context features and previous role (target/non-target) but, importantly, no interaction between these factors (please see interaction results in the main manuscript). Together, these results suggest that the effect of context and attention were additive and independent from each other. Therefore, we think that similar (especially task-relevant) context supports object stability by default rather than crucially requiring attentional selection. To tackle this issue more directly, we have addressed it in Supplementary Figure S1.

4) At the same time it is worth of notice that “attractive” interferences in working memory have already been demonstrated by an enormous amount of literature. Sometimes, when there is a change of items in working memory the end result is interference (indeed the phenomenon is dubbed “proactive interference”). But in other paradigms, when two subsequent datasets contain more similar items intrusions of past stimuli does take place. Indeed in a recent TICS review Kiyonaga et al have underlined the similarity between working memory effects and serial effects and suggested that they operate in a similar fashion. Again this pre-empties the interest on the current paper. Had the authors tackled the issue upfront, again, perhaps one would have an answer to a recent hypothesis which is receiving much attention. But the authors haven't.

As stated in the response to the reviewer's first comment, our study aimed at the putative key mechanism of serial dependence, i.e. promotion of object stability. We examined this mechanism by experimental variation of several context features in situations with several objects. We completely agree with the reviewer that the relationship between serial dependence and proactive interference represents a further important question that has not been sufficiently answered. Especially, an experimental paradigm that allows for a direct comparison between these two phenomena is still missing. However, this question was outside the scope of our study. Nevertheless, we believe that our findings provide a contribution to this topic. In particular, across our four experiments, we consistently observed that serial dependence was modulated by matching context features. As congruent context has also been repeatedly found to enhance proactive interference (Henson, 1999; Makovski & Jiang, 2008), serial dependence and proactive interference seem to behave similarly with respect to matching context. Our findings are thus in line with the assumption put forward by Kiyonaga et al. (2017) that both stem from the same underlying mechanism. In response to the reviewer's comment, we have extended our previous discussion of the review by Kiyonaga et al. (2017) starting on line 913 of the revised paper.

5) The manuscript is also not meeting entirely its own expectations. The concept of content and context in memory is presented in the introduction and even makes its way to the title. Indeed the paper promises to investigate whether context (i.e. the dimension which the experimenter will use to inform the subject upon which stimulus to judge) plays a role in serial effects. But the results are not that clear cut. In Experiment 1 where it the dimension indicating which stimulus to reproduce is color, and if the subsequent trial has the same color as the previous stimulus the effect is way larger than when the two stimuli have different colors. But in Experiment 2 the crucial dimension is “position in time” and then the extra serial effect when relating items in the same temporal position is only 20%. Does this support the authors claims? Why did they chose to study “position in time”? Why didn't they worked along the “dot-size” dimension?

We thank the reviewer for pointing this out. We apologize that we did not state clearly enough our motivation for choosing different context features such as serial position and color in the original manuscript. Our motivation was based on the assumption that serial position and color might have qualitatively different impact on serial dependence. Color represents a salient visual feature that is inherent to the visual appearance of an object. Serial position on the other hand does not have a “visual appearance” but rather arises in the temporal context of a memory episode with multiple objects. In contrast to color, serial information (together with spatial information) is considered fundamental for the creation of an object file (Kahneman et al., 1992). And, most importantly, the serial structure of the task is a source of episodic information that was expected to be used by a mechanism sub serving object continuity over time and space, as pointed out in our response to the first comment by reviewer #2.

Moreover, by using different context features such as serial position (episodic information) and color (perceptual information), we hoped to gain more insight into the processing stage at which serial dependence occurs. A purely perceptual explanation would predict a strong modulation by color, regardless of its task-relevance. In contrast, the assumption of a higher-level mechanism would predict a strong modulation by serial position. We now included a section in the Introduction (starting line 143), where we explain the choice of used context features in more detail.

While we observed that both, color and serial position modulated serial dependence in particular when they were task-relevant, we observed some differences between them. First, as pointed out by the reviewer, similar color showed a nominally higher serial dependence amplitude than similar serial position, when they were task relevant. As explained above, we did not expect the two context features to have an identical impact. However, different amplitudes might also be due to the sequential presentation in our paradigm. Previous studies (Fischer and Whitney, 2014) showed that a more recently presented stimulus causes a stronger bias. Accordingly, in our paradigm the second stimulus (previous S2) should produce a stronger bias than the first stimulus (previous S1). While this would be in line with the observed context effects when the current target was S2, the opposite is true when the current target was S1. Nonetheless, we did observe a modulation of serial dependence by congruent serial position. When assuming that other temporal effects partly counteract this effect, it is not surprising that the net effect was smaller than the effect we observed for the modulation by color.

The suggested alternative to use dot size as another context feature would also be worth investigating. We did not choose this feature in the current study because we wanted to investigate qualitatively different context features, whereas dot size would have been a visual feature similar to color. We would therefore expect effects comparable to color. Therefore, in

our newly added set of experiments we chose spatial position, as suggested by the reviewer in a later comment.

6) Another quite serious drawback of the manuscript is that several results have not been displayed clearly to the authors with graphs and are only described via the result of statistics. This makes it very difficult to gauge and absorb the effects at play. For instance the authors state that there are no interaction as for the amplitude of serial effects in Experiment 1 between the various dimensions. Does this mean that when one considers a previous stimulus which was a “non target” and had a “different color” one has a repulsive effect? Shouldn't the readers have a direct representation of all such conditions? Thus I encourage (whichever the destiny of the current submission) to display all the sub-conditions perhaps in the form of bar plots, obviously separating amplitudes and FWHM.

We thank the reviewer for pointing this out. We apologize for the low comprehensibility of our results section with regard to the interactions. We intentionally omitted a subset of the statistics for interactions in order to enhance the readability of our results section, as there is a vast number of non-significant results. We now provide all statistics for the interactions in tables in the results sections of the revised manuscript. To find a compromise between readability and completeness, we added new figures of the interactions, containing all single conditions, to the Supplemental Material (Supplementary Figure S1).

In order to address the reviewer's next point, where she/he encouraged us to incorporate Bayesian statistics, we used a different analysis approach for computing interactions in the revised manuscript. The original interaction analysis relied on the DoG curve fitting and permutation tests. However, for conditions with a non-significant effect (e.g. non-target in Experiment 1), it was unclear how to interpret the results of our original interaction analysis. To obtain easily interpretable interaction results and to provide Bayes Factors in addition to frequentist statistical values, we conducted a new interaction analysis that does not rely on the DoG curve fitting (see detailed description in new methods sections from line 296). Similar to previous approaches (Fritsche et al., 2017), we collapsed the mean response error across clockwise and counterclockwise items differences per participant and sub condition (e.g. same color, same serial position and target) and then computed the mean bias for item differences $> 0^\circ$ and $< 60^\circ$ (i.e., the bins from 10° - 50°). We took the mean bias per participant and sub-condition and then calculated a 3-way ANOVA with the factors color, serial/spatial position and previous role.

Please note that this analysis provided easily interpretable results including Bayesian statistics, even in the absence of a significant fit with the DoG model. However, after collapsing the mean response across the bins from 10° - 50° , it was no longer possible to differentiate between effects on the amplitude and the width of the bias.

As in the original interaction analysis, we observed no significant interactions between the factors across all (now four) experiments. Additionally, this new analysis again demonstrated the main effects of context features and attentional selection on the amplitude of serial dependence. We have added figures for all examined interactions in the Supplementary Figure S1 so that the data are represented in greater detail. The combination the reviewer asked for, i.e. the influence of a previous non-target with a different color, can be examined in Supplementary Figure S1a, second panel from the left. Indeed there was a small negative bias, i.e. repulsion. Therefore, we think that the inclusion of the figures helps the reader to understand the pattern of results, especially for non-targets, and therefore thank the reviewer for this valuable suggestion.

7) A related point is that some stats (like those for the interaction) are entirely missing. Also it would help to incorporate Bayesian tests. This is particularly crucial as sometimes the authors suggest there is evidence in favour of the null hypothesis (like in the aforementioned interaction)

Following the reviewer's advice, we have now included all missing statistics for the interactions. We conducted a new analysis that includes the calculation of Bayes Factors (BF), which helps to interpret the observed effects (or rather the absence of effects). For most non-significant interactions, the BFs suggest evidence for the null hypothesis, but sometimes they indicate only inconclusive evidence. This makes it is easier for the reader to interpret the evidence for an absence of interactions, which is in many cases fairly strong. Furthermore, as we report the results for the main factors of color, position and previous role in the ANOVA analyses as well, the revised manuscript now offers BFs for those main effects, too.

8) Finally the manuscript presents a rather narrow set of data and it is not trivial to generalize to other, similarly important, conditions. For instance why didn't the authors test if this pattern of results would be identical if the two items were displayed in different positions? This variable is crucial in working memory and it is important to test if it plays the same role for serial dependence. Also why didn't the authors split the data according to the sheer order of presentation in the previous trial? In other words, does the last stimulus of previous trial exert a stronger influence?

We thank the reviewer for the suggestion to include spatial position a further context feature. We have conducted two additional experiments, in which the two items were presented simultaneously and were cued via either spatial position (Experiment 3, spatial position as a task-relevant context feature) or color (Experiment 4, spatial position as a task-irrelevant context feature). We observed an impact of spatial position on serial dependence in both experiments irrespective of whether it was task-relevant. The importance of spatial position is highlighted by the observation that significant serial dependence was only present between items presented at the same spatial position. For color, we replicated the pattern of results from the first two experiments, i.e. it only affected serial dependence when it was task-relevant. We added these two experiments to the manuscript. Together, these results emphasize the role of context features for serial dependence in situations with multiple objects.

Regarding the second point, i.e. the examination of the role of stimulus order, we added a supplemental analysis for our Experiments 1 and 2 in which we split the data with respect to 1) the serial position of the previous item and 2) the serial position of the current item. To test for possible interactions, we also included the factors role of the previous items and color congruency. However, this procedure leads to a vastly reduced number of trials per condition and there was already a non-significant DoG fit for the non-target in Experiment 1 when using the whole data set. Therefore, we chose a different analysis approach by collapsing the mean response across the bins from 10°-50°, i.e. same analysis that we conducted for our new interaction calculation (see reply #6 to reviewer #2). We calculated a 4-way ANOVA with the factors previous item position (previous S1/previous S2), current item position (current S1/current S2), color (same/different) and previous role (target/non-target). For Experiment 1, no influence of either previous or current item position was evident, but the color and previous role showed significant effects. In addition, the ANOVA yielded a significant interaction between current item position and previous role. This interaction was driven by an apparently stronger influence of the previous role on a current S1 than on a current S2. For Experiment

2, we observed an effect of the current item position, i.e. S1 was generally more serially dependent on previous items than S2. The previous item position itself had no such effect. Furthermore, we found the effect of serial position congruency that we observed in the main analysis reflected by a significant interaction between the previous and the current item position and observed a significant main effect for previous role. We observed two additional interactions. First, there was a significant interaction between color and previous item position, in a way that there was a modulation by color on the serial dependence on previous S1, but not on a previous S2. While this interaction was significant, the BF indicated only inconclusive evidence. Second, we observed a significant interaction between the current item position and the previous role, as in Experiment 1 hinting at a stronger influence of the previous role for the effect on S1. Taken together, this additional analysis mainly revealed an influence of serial position of an item in the current trial, but not in the previous trial. The results indicate that the current S1 was impacted more strongly by previous targets than the current S2. This could have several reasons. On the one hand, S1 is generally closer in time to the previous trial than S2, which facilitates serial dependence (see Fischer et al., 2014). Second, S1 is maintained longer in working memory than S2, which might increase serial dependence (see Bliss et al., 2017; Papadimitrou et al., 2015). We added this additional analysis and its interpretation in the Supplementary Figure S2 and the Supplementary Tables S1 and S2. We think this information is very valuable for the understanding of the different effects in the experiments with sequential item presentation and therefore we thank the reviewer for suggesting this additional analysis.

For all these major issues I think that the current manuscript is far from ready for a premier journal like Nature Communications. Somehow the findings are potentially interesting but there is a lot to do to become a reference for future scholars.

Having said this I would like also to mention two minor points.

9) The first is that the reader should be reassured that the estimates of FWHM are independent from the amplitude estimates. How robust is their fitting procedure? Isn't it somehow a necessity that high amplitude effects can win over experimental noise and will be captured by a fitting function into their full extent whereas low amplitude effects will only be clear over a smaller range, then noise swamps the effect?

We thank the reviewer for this question, which is indeed very important for the interpretation of our results and all studies that have used the same fitting procedure by Fischer and Whitney (2014). The fitting procedure incorporates both amplitude and width as separate parameter estimates, but most studies focus their statistics on the amplitude. We investigated the FWHM estimates as well, because we think that they shed light on an interesting feature of the serial dependence bias, namely its specificity. However, we agree that one has to be careful about the interpretation of the FWHMs. They could be related to the amplitude estimate which can become evident only when both are analyzed. To test the proposed empirical dependency we used a simulation approach based on our data (see Supplementary Figure S3). We started with the effect of previous non-targets that just fell short of significance ($p = .0519$), observed in Experiment 1. To obtain realistic noise data, we subtracted the group-level serial dependence bias as fitted with the DoG method (amplitude: 0.71° , width: 0.1) from each subject's individual response error in the non-target condition of Experiment 1. Given this individual "baseline noise" a small amplitude effect as the one subtracted ($a = .71^\circ$) is just at the border of significance. Higher noise levels will therefore not lead to significant results. For simulation, we therefore reduced this baseline noise in five levels, from 100% of its original

size to 20% in steps of 20% by multiplying the individual baseline noise with the reduction factor (e.g. with 0.8 for the 80% condition). On this noise, we individually added a serial dependence bias by adding a DoG function. We used three artificial biases with different amplitudes (the original ones from the contrast, $.71^\circ$ and 2.99° as well as the mean 1.85° of both) and a fixed w value of $.065$, which is the mean of the original w estimates of this contrast. To assess the effect the amplitude has on the estimation of the width parameter, we fitted the DoG model in each of these 5 (noise level) \times 3 (amplitude) conditions and examined the relation between amplitude and width. These fittings are displayed in Supplementary Figure S3. We compared the fitted w parameters across different amplitudes and observed a significant difference between low, middle and high amplitudes (see below and Supplementary Figure S3b):

In the figure above, the darkness of the data points indicates the noise level, from high (dark color) to low (bright color). It is obvious that the noise levels affect the width estimation most strongly for the low amplitude. Therefore, we can conclude that the amplitude of the bias does affect the estimated width of a fit under a stable and especially a high noise level. While in the high amplitude condition, the w estimation is not affected by the noise level, in the low amplitude condition it approaches the “true” w with decreasing noise. However, as evident in our result, the variation in the w estimate for the low amplitude condition shows a very small range ($<1^\circ$) which would have most likely not led to a significant difference of FWHMs (the lowest FWHM difference we observed in our four experiments that led to a significant result was 8.86°). Furthermore, the case in which we put an emphasis on the modulation of the FWHM is in Experiment 1 for serial position where we could not observe an amplitude modulation but a difference in the FWHMs. As the amplitude did not differ between both conditions, and the effect of amplitude on FWHM estimates seems very small given our realistic simulation data, it seems safe to interpret the difference. Also, please note that we observed the main effects of context features on serial dependence in our newly added ANOVAs that did not depend on fitting a DoG. The only effect that did not attain significance in the ANOVAs was the width modulation by serial position observed in Experiment 1. This, however, could be expected as the ANOVA was based on the mean bias that was calculated by collapsing the response error per subject for $> 0^\circ$ and $< 60^\circ$, which should reduce sensitivity to width modulations.

Nevertheless, we agree with the reviewer that the common DoG fitting procedure should be used with caution because the relation between the fitted parameters is not entirely known.

Therefore, we hope that the inclusion of our simulation in the Supplemental Material of the paper will be of value for other scientists using the same method.

10) The second is the statement what the current findings speak against the perceptual interpretation of serial effects. Under some respect they do as they provide a case in which the mere presentation of a stimulus is not sufficient to yield an attractive effect. But at the same time this very paradigm (double presentation in the same position) is a bit peculiar and, as mentioned before, may not be that general. Even accepting that caching stimuli in working memory is crucial, still it is possible that the end effect is a warping of perception.

We fully agree that our results do not rule out that serial dependence acts on perception in general – our previous discussion rather intended to point out that an exclusively perceptual interpretation of our results is not possible. The results we observed suggest an origin of serial dependence in a memory representation or a decision. In our new experiments, we observed a modulation by context for the more commonly used simultaneous presentation of two items at different spatial positions. Still, we observed a strong modulation of serial dependence by the previous role of an item. But, as the reviewer points out, it is still possible that serial dependence affects the perception of an upcoming stimulus, as our study is not designed to rule that out. Therefore, the present results give new insights into the origin of serial dependence, whereas they are not conclusive concerning which type of representation it affects. As already acknowledged in our responses to comments 2 and 3 by reviewer #1, the previous version of the Discussion was kept too simple. It was now rewritten (starting line 959) to include more nuanced in our statements.

****REVIEWERS' COMMENTS:**

Reviewer #1 (Remarks to the Author):

The authors have done a commendable job with the new experiments and revised text. The manuscript is greatly improved. I only have a couple of additional suggestions.

1. Line 886. "Interestingly, the impact of spatial position as a context feature on serial dependence has remained unclear given the existing literature." This is not accurate and the entire section on spatial tuning should be rephrased: the overwhelming evidence is that serial dependence is, in fact, spatially tuned. In fact, the authors' own results in the present manuscript show spatial tuning of serial dependence. There are several additional papers that show spatial tuning of serial dependence (all of these should be cited in the discussion section): Collins, T., 2019, Scientific Reports "The perceptual continuity field is retinotopic"; Manassi, et al., 2019, Scientific Reports, "serial dependence in a simulated clinical visual search task"; Cicchini, et al., 2017; Fischer & Whitney, 2014. There are therefore 5 (!) papers, including the current manuscript, that show spatial tuning of serial dependence. The most recent paper is the one by Collins, and is the most thorough paper to date on the spatial tuning of serial dependence. Whether the spatial tuning is entirely retinotopic or not is immaterial for the purpose of this discussion section, because every paper but one (Fritsche et al) agrees that serial dependence is spatially tuned. The reference to Fritsche et al 2017 (their null result) should be dropped, as it is inconclusive (it does not prove the absence of spatial tuning, it does not cast doubt on the 5 sets of results cited above, it is simply a null result). Instead, the manuscript text should focus on the 4 other papers that have shown spatially tuned serial dependence, and how the present results are consistent with those findings. The whole section needs to be revised because it ignores the substantial existing proof for spatial tuning (e.g., modify the unsupported statement that "...even though location might be a crucial feature for an object, its importance might arise in particular when multiple items are processed..." Location is important for serial dependence in many studies and the current results agree with that.

2. 2. Supplemental Fig 3 is interesting, but the effect of noise is actually very very small. The Y axis range is expanded, which exaggerates the apparent effect size. Also, the result is "significant" only if not corrected. However, those multiple tests need to be Bonferroni corrected (or at least FDR corrected). In any case, the actual effect size is very small. Graphing a Y axis that includes zero would give a more balanced perspective on the effect of noise. The authors should consider rephrasing the overly strong statement that "Based on this finding we concluded that the amplitude did affect the width estimation at a high noise level" to something like "the amplitude may have a slight effect on the width estimation".

Reviewer #2 (Remarks to the Author):

Dear all,

firstly let me say that I have appreciated much the introduction of new experiments and all the work done. Now the work clearly more thorough and most of the points I have raised (in general complaining of the fact that the paper was not focussed) are now addressed.

I would only suggest to invert the order between exp 3 and 4 (color cue) as they would match exactly those of exp 1 (color cue) and 2.

Also I have some suggestions which are no pre-requisite for publication. As personal taste I would have outset with the experiments on space as these resemble most the typical conditions used in serial dependence experiments (see Fisher and Whintey). Also I would like to see somehow Figure S1 in the main manuscript. I find it particularly valuable as it helps assessing at a glance all the effects and lack of interactions mentioned in the tables. But I leave it up to the authors.

On the other hand I found that the caption of Figure S1 is somewhat not polished. I would not state that they are meant to show the lack of interaction (as they also show the main effect). If this change is accepted then probably the first 4 lines are unnecessary.

For the rest I am happy with the current ms.

We are grateful to the reviewers for their enthusiasm about our revised manuscript and their suggestions for improvement. We have addressed each of their points by rephrasing our discussion, editing the Supplemental Materials and reversing the order of Experiments 3 and 4.

Reviewer #1 (Remarks to the Author):

The authors have done a commendable job with the new experiments and revised text. The manuscript is greatly improved. I only have a couple of additional suggestions.

1. Line 886. “Interestingly, the impact of spatial position as a context feature on serial dependence has remained unclear given the existing literature.” This is not accurate and the entire section on spatial tuning should be rephrased: the overwhelming evidence is that serial dependence is, in fact, spatially tuned. In fact, the authors’ own results in the present manuscript show spatial tuning of serial dependence. There are several additional papers that show spatial tuning of serial dependence (all of these should be cited in the discussion section): Collins, T., 2019, Scientific Reports “The perceptual continuity field is retinotopic”; Manassi, et al., 2019, Scientific Reports, “serial dependence in a simulated clinical visual search task”; Cicchini, et al., 2017; Fischer & Whitney, 2014. There are therefore 5 (!) papers, including the current manuscript, that show spatial tuning of serial dependence. The most recent paper is the one by Collins, and is the most thorough paper to date on the spatial tuning of serial dependence. Whether the spatial tuning is entirely retinotopic or not is immaterial for the purpose of this discussion section, because every paper but one (Fritsche et al) agrees that serial dependence is spatially tuned. The reference to Fritsche et al 2017 (their null result) should be dropped, as it is inconclusive (it does not prove the absence of spatial tuning, it does not cast doubt on the 5 sets of results cited above, it is simply a null result). Instead, the manuscript text should focus on the 4 other papers that have shown spatially tuned serial dependence, and how the present results are consistent with those findings. The whole section needs to be revised because it ignores the substantial existing proof for spatial tuning (e.g., modify the unsupported statement that “...even though location might be a crucial feature for an object, its importance might arise in particular when multiple items are processed...” Location is important for serial dependence in many studies and the current results agree with that.

We agree with the reviewer that the discussion section about spatial tuning was not well balanced. We have included the suggested references and reframed and shortened (due to space limitations) the original section to: “The observed influence of spatial position is in line with previous studies demonstrating a spatial tuning of serial dependence (Cicchini et al., 2017; Collins, 2019; Fischer & Whitney, 2014; Manassi et al., 2019).”.

2. 2. Supplemental Fig 3 is interesting, but the effect of noise is actually very very small. The Y axis range is expanded, which exaggerates the apparent effect size. Also, the result is “significant” only if not corrected. However, those multiple tests need to be Bonferroni corrected (or at least FDR corrected). In any case, the actual effect size is very small. Graphing a Y axis that includes zero would give a more balanced perspective on the effect of noise. The authors should consider rephrasing the overly strong statement that “Based on this finding we concluded that the amplitude did affect the width estimation at a high noise level” to something like “the amplitude may have a slight effect on the width estimation”.

Thank you for pointing this out. The noise effect was indeed very small. We have changed the figure as suggested by the reviewer. In addition, we now display the data in a “zoom-in” panel, to make the slight noise effect visible for the reader. We now also report the Bonferroni-corrected p-values and have rephrased the overly strong statement about the effect accordingly.

Reviewer #2 (Remarks to the Author):

Dear all,

firstly let me say that I have appreciated much the introduction of new experiments and all the work done. Now the work clearly more thorough and most of the points I have raised (in general complaining of the fact that the paper was not focussed) are now addressed.

I would only suggest to invert the order between exp 3 and 4 (color cue) as they would match exactly those of exp 1 (color cue) and 2.

We appreciate this suggestion and have changed the order of the Experiments accordingly. Experiment 3 now describes the color cue manipulation and Experiment 4 the spatial cue.

Also I have some suggestions which are no pre-requisite for publication. As personal taste I would have outset with the experiments on space as these resemble most the typical conditions used in serial dependence experiments (see Fisher and Whintey). Also I would like to see somehow Figure S1 in the main manuscript. I find it particularly valuable as it helps assessing at a glance all the effects and lack of interactions mentioned in the tables. But I leave it up to the authors.

We thank the reviewer for these valuable suggestions. We also agree with the reviewer that the experiments with simultaneous item presentation (Experiments 3 and 4) would represent a good start point of our study as they refer to a more common condition in visual science. However, we have decided to keep the original order that actually reflects the order in which the study was originally conceived. We also appreciate the idea of incorporating Figure S1 into the main manuscript. However, as we already had to shorten the manuscript to comply with the format requests of the journal, we have pooled all interaction results and Figures and moved them to the Supplementary material. In this way, all the information about the interactions effects are in one place and can be assessed at a glance.

On the other hand I found that the caption of Figure S1 is somewhat not polished. I would not state that they are meant to show the lack of interaction (as they also show the main effect). If this change is accepted then probably the first 4 lines are unnecessary.

We agree with this point and have changed the general description of this figure to “This figure illustrates the relationship between context features and the previous role of an item for all four experiments.” and also removed the last four lines as suggested.

For the rest I am happy with the current ms.